# A semi-synthetic regulon enables rapid growth of yeast on xylose

Venkatesh Endalur Gopinarayanan[1] & Nikhil U. Nair [1]

Nutrient assimilation is the first step that allows biological systems to proliferate and produce value-added products. Yet, implementation of heterologous catabolic pathways has so far relied on constitutive gene expression without consideration for global regulatory systems that may enhance nutrient assimilation and cell growth. In contrast, natural systems prefer nutrient-responsive gene regulation (called regulons) that control multiple cellular functions necessary for cell survival and growth. Here, in *Saccharomyces cerevisiae*, by partially- and fully uncoupling galactose (GAL)-responsive regulation and metabolism, we demonstrate the significant growth benefits conferred by the GAL regulon. Next, by adapting the various aspects of the GAL regulon for a non-native nutrient, xylose, we build a semi-synthetic regulon that exhibits higher growth rate, better nutrient consumption, and improved growth fitness compared to the traditional and ubiquitous constitutive expression strategy. This work provides an elegant paradigm to integrate non-native nutrient catabolism with native, global cellular responses to support fast growth.

[1] Department of Chemical and Biological Engineering, Tufts University, Medford, MA 02155, USA. Correspondence and requests for materials should be addressed to N.U.N. (email: Nikhil.Nair@tufts.edu)

Efforts in synthetic biology and metabolic engineering have largely focused on rationally designing regulatory infrastructures around biosynthetic/anabolic pathways. In recent years, dynamic pathway regulation has yielded significant improvements in product titers using either native[1,2] or heterologous transcription factors[3–5]. Conversely, rationally designed regulatory controls for efficient and complete utilization of exogenously available nutrients in synthetic biological systems are underdeveloped. Current efforts to engineer nutrient assimilation pathways take a straightforward approach of over expressing catabolic pathway enzymes without regard for how that integrates into the larger cellular infrastructure that encompasses central metabolism, stress-responses, cell doubling, etc. Examples include engineering pentose catabolism in *S. cerevisiae*[6–13], C1 (viz. $CO_2$ or methanol) feedstock usage in *E. coli*[14,15], or even amorphous cellulose utilization by various yeasts and bacteria[16–18]. In stark contrast, natural systems often use genome-scale regulatory infrastructures, called regulons, to coordinate nutrient catabolism with other cellular functions[19–26]. Such systems include sensors to detect a specific nutrient (input), signal transduction and integration (computation), and global genetic regulation (output). One of the best studied native regulons is the galactose (GAL) system in the yeast *Saccharomyces cerevisiae* (*S. cerevisiae*), where Gal3p-mediated sugar detection initiates a genome-wide response effected by Gal4p and Gal4p-responsive transcription factors (TFs)[25].

In this study, we first assess the role of the GAL regulon in enabling efficient galactose utilization for cell growth by decoupling its regulatory responses from sugar catabolism. We provide evidence that regulon-controlled galactose assimilation is better than constitutive expression of the catabolic genes in supporting fast growth rates to higher cell densities. Next, we assess whether a regulon could enable more complete and efficient utilization of a nutrient that is non-native to this yeast—xylose. We first adapt the GAL regulon to respond to xylose through directed evolution of Gal3p, coupling nutrient stimulus with sensing, computation, and regulatory actuation. Next, by using a rational, model-guided approach, we test two different positive feedback signal transduction loop designs for the regulon and demonstrate their individual merits and weaknesses. We also show that implementation of a GAL-type xylose-responsive regulon can regulate multiple genes across the yeast genome and enable more homogeneous population-wide gene expression. By integrating a minimal set of heterologous catabolic genes into the synthetic regulon we demonstrate high cellular growth rates and high final cell densities on xylose as well as better growth in non-inducing carbon sources. Finally, we compare the genome-wide expression profiles of strains grown with regulon assistance and conventionally engineered strains to identify mechanistic reasons that account for the different phenotypes observed. We posit that this study strongly supports the need to re-evaluate how nutrient assimilation systems are currently implemented and introduces a paradigm of adapting a native regulon for efficient non-native sugar assimilation.

## Results

**Downstream GAL regulon genes support fast growth.** The GAL regulon exerts control over the initial galactose metabolic genes (Leloir pathway) as well as several downstream genes not directly involved in assimilating galactose[27–29]. While the roles of initial galactose catabolic and regulatory genes have been established, the control exerted by the regulon on downstream genes and their phenotypic effects are not well elucidated. We decided to assess the role played by downstream genes of the GAL regulon in growth on galactose by fully- or partially uncoupling their

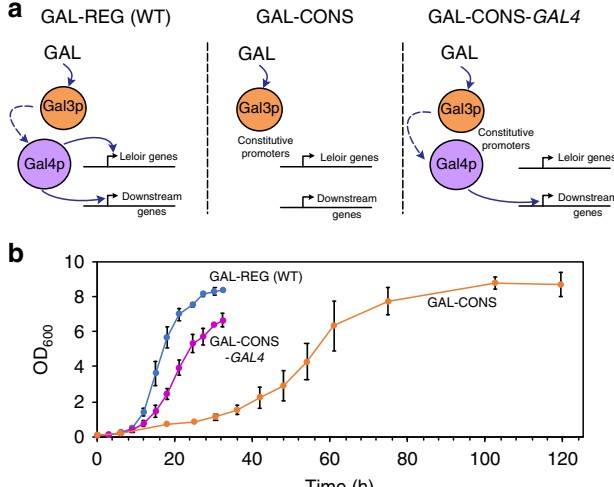

**Fig. 1** Gal4p-mediated activation of genes other than the Leloir pathway enzymes support fast growth on galactose. **a** Schematic of regulon design used to test effects of regulon on growth on galactose. The left panel denotes the wild-type with intact regulon that has Leloir pathway genes and downstream genes controlled by Gal4p. The middle panel represents the GAL-CONS strain with Gal4p knocked out and Leloir pathway genes under strong constitutive promoters. The right panel denotes the GAL-CONS-*GAL4* mutant with downstream genes under Gal4p, but Leloir pathway genes under constitutive promoters. **b** Growth of the three strains in galactose. Each of the data points denotes the average of three individual replicates, error bars are ± sd

communication. To decouple growth on galactose from the regulon's function, we knocked out *GAL4*, the gene that encodes for the master GAL regulon transcription factor and placed the initial galactose metabolic genes (*GAL1*, *GAL7*, and *GAL10*) under the control of strong constitutive promoters, *TEF1p*, *GPM1p*, and *TPI1p*, respectively, to create a constitutive galactose metabolic gene expression strain, GAL-CONS (Fig. 1a). We compared the growth rates and final cell densities of GAL-CONS with GAL-REG (wild-type) and observed that the former had a > 5-fold lower growth rate and took three-times longer to reach stationary phase (Fig. 1b). The decrease in growth rate can either be attributed to inability of the strain GAL-CONS to activate the downstream genes of GAL regulon that are required for growth, or the difference in promoter strengths between the GAL and constitutive promoters that transcribe the Leloir pathway genes, or both. To determine true cause of growth rate decrease, we re-introduced Gal4p in GAL-CONS but deleted genomic Leloir pathway genes (and placed the genes under constitutive expression) as well as *GAL3* and *GRE3* (which encodes for non-specific aldose reductase) to create the strain GAL-CONS-*GAL4* (Fig. 1a). This re-factored, partially coupled system should enable activation of downstream genes through Gal1p-Gal80p-Gal4p pathway[30], but keeps the Leloir pathway genes out of the GAL regulon control. Thus, if the downstream genes of the GAL regulon assist in growth, the partially coupled strain should have growth rates higher than the GAL-CONS strain. On the other hand, if the observed decrease is a result of the difference in promoter strengths between native GAL promoters and constitutive promoters, the GAL-CONS-*GAL4* strain should have the same growth rate as that of the GAL-CONS strain. We tested the GAL-CONS-*GAL4* strain for growth in galactose and observed that the strain recovers a significant portion of its growth fitness relative to GAL-CONS (Fig. 1b) suggesting that the downstream genes under the control of the regulon *trans*-activated by Gal4p positively affect the ability of yeast to grow on galactose. It should be

noted that constitutive promoters (expressed on multicopy plasmids) have higher expression strength than GAL promoters (Supplementary Fig. 1 and Supplementary Note 1), thereby suggesting the promoter strengths do not play a major role in dictating the growth rate of the strains tested. This can be seen when comparing growth of wild-type and GAL-CONS-GAL4 strains (Fig. 1b). Here, the benefits of using GAL promoters are demonstrated by the slightly higher growth rate. However, most of growth benefit is gained by activation of downstream (non-Leloir) genes. Next, we wanted to test whether this observation can be extended to alternative nutrients, particularly to those that are non-native to yeast.

**Design of a synthetic GAL-type xylose regulon.** So far, implementation of heterologous sugar assimilation systems in yeast and bacteria have relied on constitutive overexpression of metabolic genes[6,8,9,14,15,31,32] rather than regulon-controlled sugar assimilation. This is not only because there is no evidence to demonstrate the advantages of regulon-assisted growth, but also due to lack of tools and guidelines available to build a synthetic regulatory network that can coordinate nutrient detection with metabolism and cell growth. Hence, rather than building a synthetic regulatory network from the bottom-up where each of the genes to be activated are chosen rationally to build an elaborate regulatory and metabolic network, we decided on a more efficient approach by reverse-engineering the GAL regulon into a xylose-responsive regulon. We hypothesized that since many of the downstream genes required for growth are relatively conserved irrespective of carbon source, most of the genes activated by the GAL regulon would also be beneficial for growth on xylose. To realize this regulatory system, we decided to engineer the three different components of the regulon—nutrient detection, signal transduction, and nutrient metabolism (Fig. 2). First, the galactose sensor Gal3p must be adapted to detect xylose and bind Gal80p to activate the regulon. Second, the signal transduction loop present in the galactose regulon must be re-designed for the xylose regulon, and finally, the genes required for xylose metabolism must be placed under the control of the regulon.

**Engineering Gal3p for improved response to xylose.** To identify mutations within Gal3p that enhance its responsiveness to xylose, we developed a reporter strain and a robust selection and screening system. We deleted the genes of galactose metabolism (GAL1, GAL7, and GAL10) so that galactose acts only as an inducer of GAL regulon and like xylose, is not metabolized. We also knocked out GAL3, to be expressed through a plasmid for mutagenesis, and GRE3 (non-specific aldose reductase), to prevent reduction of sugars to polyols, to create the reporter strain, VEG16. Thus, the reporter strain lacks both the sensory proteins as well as metabolic enzymes required for galactose metabolism. Next, we developed a selection and screen, based on G418 antibiotic resistance and enhanced green fluorescent protein (EGFP), respectively, by placing the two marker genes KANMX and EGFP under the control of bidirectional GAL1p and GAL10p promoters. We placed GAL3 downstream of its own native promoter, GAL3p, along with the selection and screening construct into a multicopy plasmid (pVEG8). Thus, any Gal3p–sugar interaction would activate the GAL regulon, resulting in expression of the KANMX gene for high-throughput antibiotic selection and EGFP for quantitative fluorescence screening (Fig. 3a). Our initial fluorescence screening assays with Gal3p-WT showed a typical dose–response sigmoidal curve with galactose, as expected. When tested with xylose, we observed a linear increase in fluorescence at xylose concentrations above 2% (Fig. 3b). When compared to its native substrate galactose, the fluorescence exhibited in the presence of xylose was

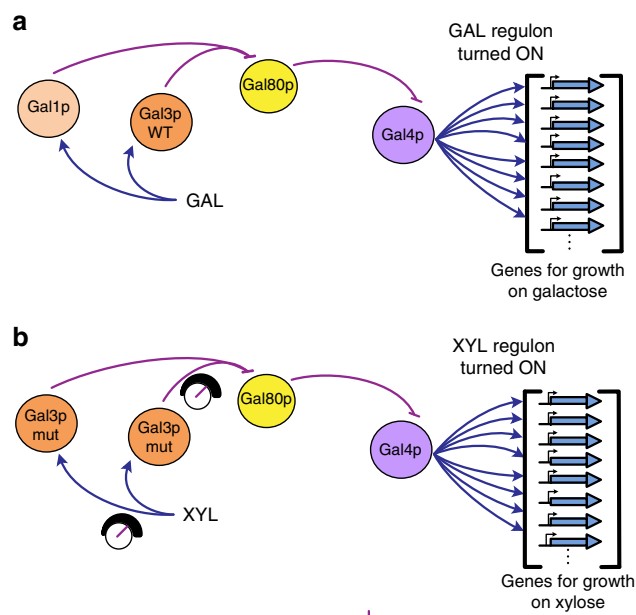

**Fig. 2** Design of a semi-synthetic GAL-type xylose regulon. **a** Schematic of galactose-based activation of the GAL regulon, where galactose-bound Gal3p relieves repression of Gal4p by binding with Gal80p, thereby turning ON the regulon, including the genes required for growth on galactose. Gal1p, one of the GAL regulon genes also interacts with Gal80p creating a dual-positive feedback loop. **b** Design of the xylose regulon. The first stage involves protein engineering of Gal3p such that Gal3p–xylose interaction relieves repression on Gal4p. The second stage involves capturing the dual-positive feedback loops created by Gal3p and Gal1p by expressing a Gal3p mutant under two different promoters. The final stage involves integrating genes required for growth on xylose under GAL activated promoters to create a xylose sensing and metabolizing semi-synthetic regulon

several-fold lower and observable only at high-sugar concentrations where transport is not expected to be an issue[33,34]. But the presence of fluorescence at high concentrations also indicates weak Gal3p–xylose interaction, suggesting that Gal3p active site is sufficiently flexible to accommodate xylose. Next, we carried out four sequential rounds of mutagenesis on Gal3p and screened for mutants that exhibit increased fluorescence as well as increased fold-change (fluorescence in the presence of xylose over fluorescence in the absence of xylose) (Supplementary Note 2, Supplementary Fig. 2 and Supplementary Table 1). The final mutant obtained, Gal3p-Syn4.1, exhibited a 16-fold increase in fluorescence with increased sensitivity (even at 0.2% xylose) and lower background fluorescence (Fig. 3c). Comparing fold changes at 2% xylose of Gal3p-Syn4.1 with fluorescence of Gal3p-WT at 2% galactose, we observed that the mutant exhibits a similar fold-change, suggesting that Gal3p-Syn4.1 has sufficient transcriptional activation strength. We also checked the fluorescence profile of the best Gal3p variants from every round of mutagenesis in the presence of galactose and observed a 100-fold increase in sensitivity when compared to Gal3p-WT (Fig. 3d). Thus, by targeting mutations at the protein–protein interaction sites and carrying out random mutagenesis on the entire protein, we obtained a multi-sugar sensor with more than 15-fold induction in the presence of xylose, while still retaining its native galactose-binding function.

**Model-assisted tuning of XYL regulon activation and control.** The GAL regulon has been described mathematically as a bi-stable system with negative- and positive-feedback loops[35].

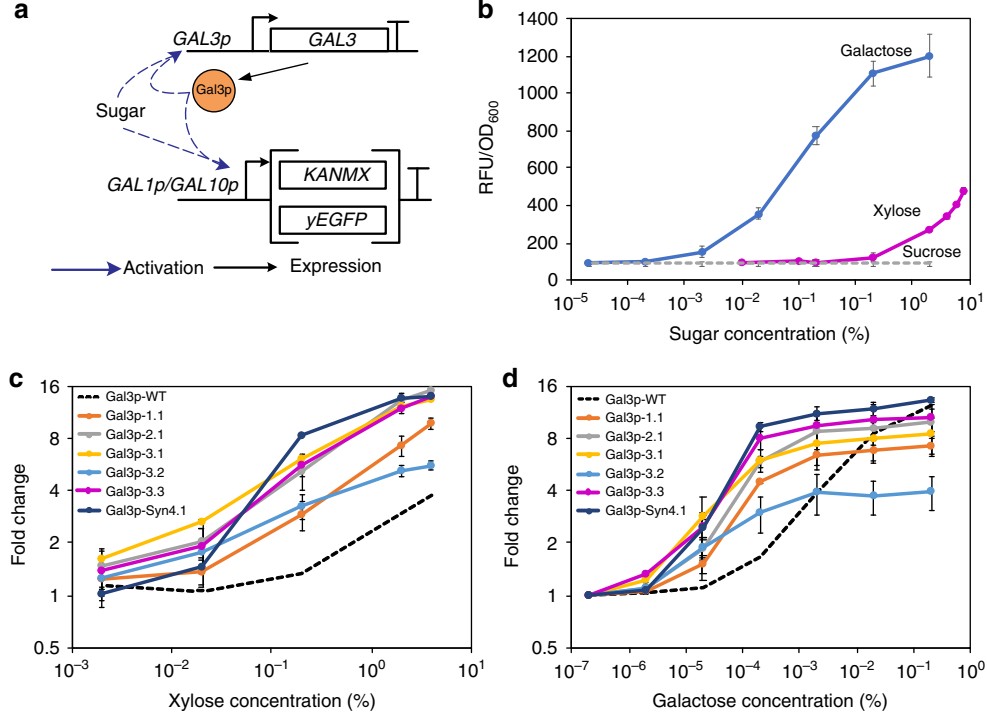

**Fig. 3** Engineering Gal3p to respond to xylose. **a** Design of the screen and selection systems. Gal3p expressed under its native promoter, in the presence of activation sugar (galactose/xylose), would activate the GAL regulon and switch ON *GAL1p* and *GAL10p* to drive expression of *KANMX* and *EGFP*, resulting in G418 sulfate resistance (selection) and fluorescence (screen). **b** Gal3p–WT interaction with galactose and xylose measured using the fluorescence assay. VEG16 transformed with selection and screening construct was grown in sucrose with varying concentrations of galactose ($2\% - 2 \times 10^{-6}\%$) or xylose ($8\% - 2 \times 10^{-3}\%$). **c** Fold-change in fluorescence of the best Gal3p mutants when incubated with varying concentrations of xylose. **d** Fold-change in fluorescence of the best Gal3p mutants when incubated with varying concentrations of galactose. Each data point represents average of three individual replicates ± sd

The negative feedback is mediated by the repressive function of Gal80p on Gal4p, whereas the positive feedbacks are mediated through Gal3p and Gal1p-based de-repression of Gal4p[30,36]. Further, the high-basal, weak, *GAL3p* promoter-driven *GAL3* expression along with a low-basal, strong, *GAL1p* promoter-driven *GAL1* expression creates a dual-positive feedback loop, which has been shown to increase sensitivity and decrease noise[30,35,37]. Such a dual-positive feedback loop enables rapid, strong, and homogeneous expression during induction and low-basal expression in the absence of inducer. To recapitulate this, we decided to build a dual-positive feedback loop. However, since Gal1p is a galactokinase with affinity towards galactose rather than xylose, we could not use Gal1p to create the dual-positive feedback loop. Instead, we hypothesized that placing *GAL3-Syn4.1* under both, *GAL3p* and *GAL1p* promoters should be sufficient to create a dual-positive feedback loop. To test this, we used the ODE model from Venturelli et al.[35], that captures the interplay between Gal3p, Gal80p, Gal4p, and Gal1p and modified it by modeling GFP expression under *GAL1/10p* promoter so that it can be compared with our experimental results (Supplementary Note 3). We assessed the cooperativity of Gal4p binding on *GAL10p*, *GAL3p,* and *GAL80p* promoters used in the model experimentally by expressing EGFP under the three promoters. We measured fluorescence output at varying concentrations of xylose, which was then used to fit a Hill curve to determine cooperativity (Supplementary Fig. 3).

By varying the rate of galactose input for the two feedback models, we show that the sensitivity factor (concentration of inducer required to attain half the maximum fluorescence) is lower for the dual feedback than for the single feedback system (Supplementary Fig. 4A). We carried out the experiment and

observed that the trend of the simulation matched with experimental data (Supplementary Fig. 4A, B). While it has been known that the dual feedback system created by Gal3p and Gal1p increases sensitivity when compared to single feedback system without Gal1p, we show that simply by having *GAL3* under *GAL3p* and *GAL1p* promoters, we can achieve similar sensitivity.

In the case of xylose regulon, Gal3p-Syn4.1–xylose–Gal80p interaction is much weaker than Gal3p-WT–galactose–Gal80p interactions (Fig. 3c, d). To take that into consideration, we varied the forward-binding rate constant (kf83) of Gal3p binding to Ga80p in the presence of inducer (galactose/xylose) over five orders of magnitude from 0.1–10,000 nM/min and tracked the sensitivity factor. We show that at very low kf83 values, single and dual feedback systems display similar sensitivity factor, probably due to poor association of Gal3p-Gal80p (Supplementary Fig. 4C and Fig. 4b). As we increased kf83, at intermediate strengths of Gal3p-Gal80p binding, sensitivity factor is significantly lowered for the dual feedback loop (Supplementary Fig. 4E and Fig. 4b). Finally, when Gal3p–Gal80p interaction is strong, the sensitivity factor saturates in both the feedback models with the dual feedback having higher sensitivity than the single feedback system (Supplementary Fig. 4A and Fig. 4b). Thus, the model predicts that dual feedback loop is more effective when Gal3p has relatively weaker interactions with the inducer, as is the case with xylose regulon. To test the model prediction that increased sensitivity could be observed under xylose induction, we placed *GAL10p* upstream of *EGFP* and compared fluorescence for strains with single and dual-positive feedback. We incubated them with different concentrations of xylose or galactose with either Gal3p-WT or Gal3p-mut (Supplementary Fig. 4B, D, F) and observed that simulation and experiments have similar trends (Fig. 4c and

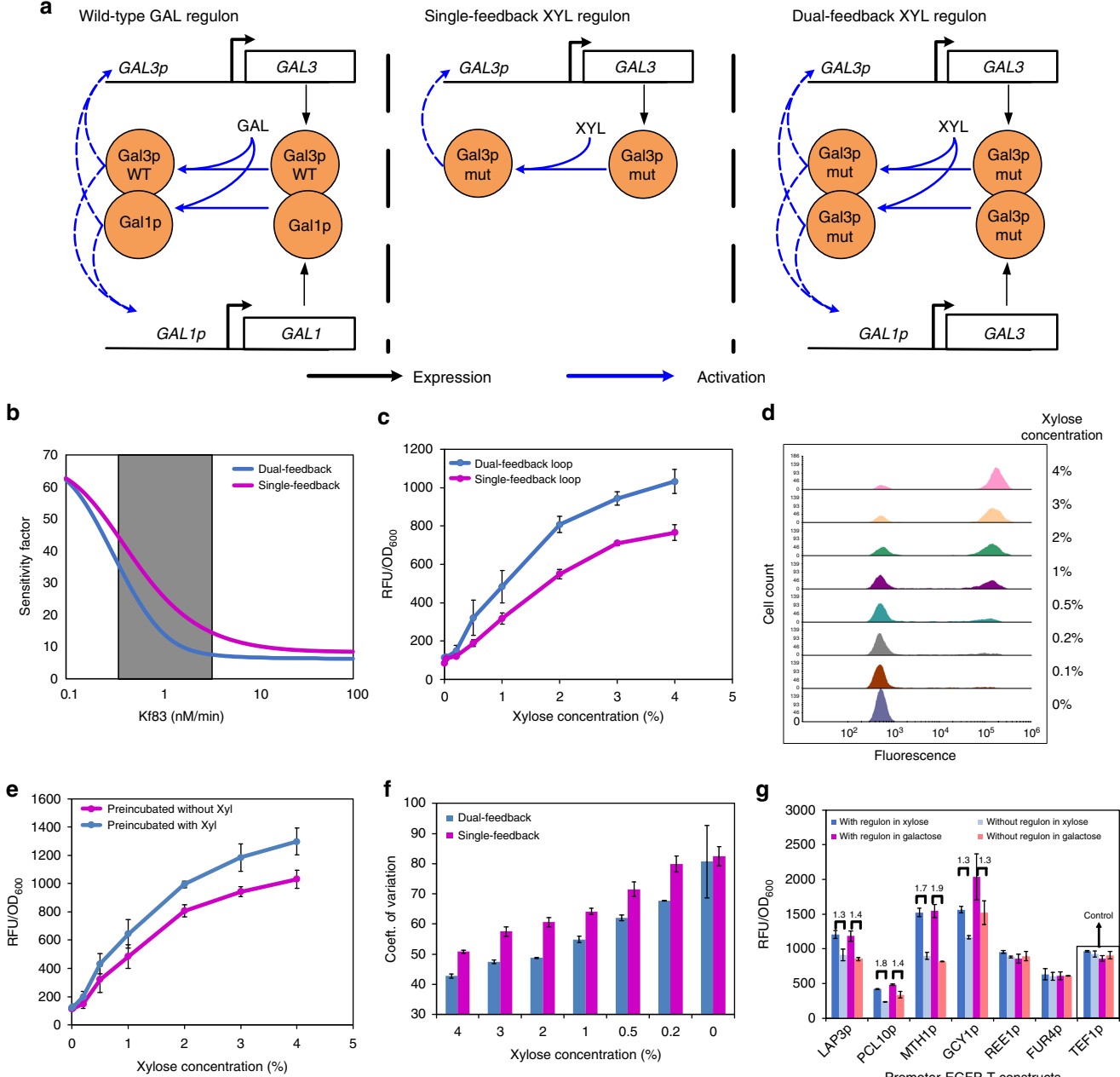

**Fig. 4** Tuning of xylose regulon activation. **a** Schematic of dual-positive feedback loop in the galactose regulon (left panel), single- (middle panel), and dual-positive feedback loop (right panel) in the xylose regulon. While Gal3p and Gal1p act in concert to create the dual feedback in the galactose regulon, Gal3p-Syn4.1 driven by two different promoters is used to create the dual-positive feedback loop in the xylose regulon. **b** Model simulation of sensitivity factor (concentration of inducer required to attain half of the maximum fluorescence) vs. forward-binding rate constant of Gal3p-Gal80p (kf83). The shaded region represents the range of kf83 where the difference between the two feedback models is prominent. **c** Fluorescence under *GAL10p* promoter in single and dual feedback systems at different concentrations of xylose (4, 3, 2, 1, 0.5, 0.2, 0.1, 0%). **d** Flow cytometry histograms of cell populations from dual feedback system that are either ON or OFF at different xylose concentrations, showing bimodal distribution of cells when induced with xylose. **e** Fluorescence of EGFP gene under *GAL10p* promoter under dual feedback system at different concentration of xylose either pre-incubated in 4% xylose or 0% xylose shows hysteresis. **f** Coefficient of variation (CV) obtained from flow cytometry measurements of single and dual feedback systems as a measure of cellular heterogeneity when induced with different concentrations of xylose. **g** Promoters that drive expression of downstream genes of the galactose regulon were used to express *EGFP*. Fluorescence was measured for cells grown on ethanol/glycerol in the presence or absence of xylose and galactose regulon. The fold-change obtained in xylose and galactose regulons are shown on top of the bars for comparison. Each data point represents average of three individual replicates ± sd

Supplementary Fig. 4). In Fig. 4c, we can see that the dual-positive feedback loop shows increased fluorescence than a single feedback loop. We observed a similar increase in sensitivity with other GAL promoters—*GAL1p* and *GAL7p* (Supplementary Fig. 5).

The GAL regulon has been known to exhibit bimodality (results in heterogeneous population in suboptimal environment, thus increasing fitness) and hysteresis (a history dependent response to galactose), which are characteristic features of a bi-stable system[35]. To test if the xylose activated regulon still retains

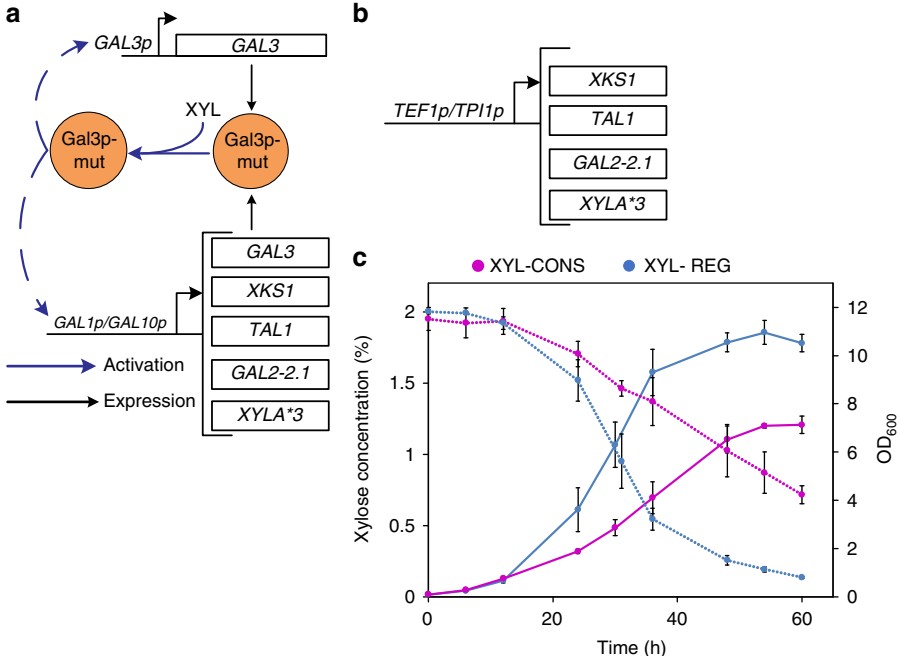

**Fig. 5** Growth on xylose facilitated by the xylose regulon. **a** Design of the genes placed under the semi-synthetic xylose regulon and its mode of activation. Apart from *GAL3-Syn4.1* placed under *GAL3p* and *GAL1p* promoters, other metabolic genes *XYLA*3*, *XKS1*, *TAL1*, and xylose transporter *GAL2-2.1* were expressed under *GAL1p* and *GAL10p* promoters. **b** Design of the metabolic control strain (XYL-CONS) built by placing the genes *XYLA*3*, *XKS1*, *TAL1*, and *GAL2-2.1* under strong *TEF1p* and *TPI1p* promoters. **c** Growth (bold lines) and xylose consumption profile (dotted lines) of the engineered strain, XYL-REG, and metabolic control, XYL-CONS under aerobic conditions. Each data point represents average of three individual replicates ± sd

bistability observed in the parent regulon, we decided to test bimodality and hysteresis in the XYL regulon. To demonstrate bimodality, we integrated *GAL10p-EGFP-T* cassette into the chromosome and compared fluorescence of the two feedback systems at a cellular level under different concentrations of xylose using flow cytometry. Over the concentration range tested, we observed distinct populations of cells that were either turned ON or OFF confirming that the xylose regulon still retains bimodality (Fig. 4d). Next, we pre-incubated the yeast strain carrying dual feedback system in media with and without the inducer (xylose) for 24 h, and later shifted the cells to media with varying concentrations of xylose. We observed a pre-incubation-dependent response (hysteresis) at all concentrations of xylose tested. The cells that were pre-incubated with xylose showed a higher fluorescence than strains that were not incubated with xylose (Fig. 4e). Altogether, these data show that the GAL-type xylose regulon retains bistability observed in the galactose regulon[35]. In 2005, Brandman et al.[38] hypothesized using mathematical simulations that interlinked dual-positive feedback loops with fast and slow feedback responses, results in a faster response as well as a more stable signal output with low noise when compared to using either of the feedback loops in isolation. The fast response was attributed to the fast feedback loop and the low noise with a stable response was attributed to the slow feedback loop. Since *GAL3p* and *GAL1p* promoters provide fast and slow feedback loops, respectively, we decided to test if the absence of slow feedback loop (*GAL1p*) would result in increased noise in the system. To test that, we calculated coefficient of variation (CV), a measure of cellular heterogeneity and noise, for dual feedback and single feedback systems. The dual feedback loop had lower CV than the single feedback loop across different concentrations of xylose tested (Fig. 4f), consistent with observations of Brandman et al.

Finally, we also show that the XYL regulon activates downstream genes such as *GCY1*, *LAP3*, *MTH1*, and *PCL10* that have been known to be activated by the GAL regulon (Fig. 4g, Supplementary Note 4, Supplementary Fig. 6).

**A semi-synthetic XYL regulon enables better growth on xylose.** We placed genes necessary for xylose metabolism (*XYLA*3*[39], *XKS1*, *GAL2-2.1*[40], and *TAL1*) under the control of *GAL1p* and *GAL10p* promoters and transformed them along with the dual feedback loop system to create the strain, XYL-REG that has a complete xylose regulon capable of xylose detection and metabolism (Fig. 5a). We built a metabolic control strain, XYL-CONS, where the four xylose catabolic genes were placed under the control of strong constitutive *TEF1p* and *TPI1p* promoters (Fig. 5b). Initial growth studies in synthetic complete (SC) xylose medium resulted in a growth rate of 0.12 per hour for with XYL-REG and 0.07 per hour for XYL-CONS (Supplementary Fig. 8A). However, xylose was not fully consumed possibly due to nutrient limitation in the minimal SC medium. Hence, we tested growth in complex YP medium supplemented with 2% xylose and observed growth rates of 0.15 per hour for XYL-REG and 0.06 per hour for XYL-CONS. While XYL-REG reached an $OD_{600}$ of 11, the XYL-CONS attained an $OD_{600}$ of only 7. In concurrence with the OD values, most of xylose was consumed by XYL-REG and less than 0.15% residual xylose was observed at the end of 60 h. However, in the case of XYL-CONS, almost one-third (>0.7%) of xylose remained unused in the spent medium (Fig. 5c). These data show that implementation of a synthetic xylose regulon resulted in a higher growth rate, complete xylose consumption, as well as increased biomass density with minimal metabolic engineering.

We also compared the effect of other GAL regulon designs on growth on xylose. We have shown that dual-positive feedback exhibits better sensitivity and lower noise when compared to single feedback design. Next, we tested if these characteristics would also translate to improved growth. We transformed plasmids carrying necessary genes for xylose metabolism under

| Table 1 Comparison of growth rates ($\mu$) for different strains cultured in different sugars—xylose, galactose, sucrose, and glucose | | | | |
|---|---|---|---|---|
| **Strain** | **Growth rate on sugar, $\mu$ (per hour)** | | | |
| | **Xylose** | **Galactose** | **Sucrose** | **Glucose** |
| WT (GAL-REG) | – | 0.30 ± 0.02 | 0.30 ± 0.01 | 0.31 ± 0.01 |
| XYL-CONS | 0.06 ± 0.01 | – | 0.24 ± 0.01 | 0.25 ± 0.01 |
| XYL-REG | 0.15 ± 0.01 | – | 0.31 ± 0.01 | 0.30 ± 0.002 |
| GAL-CONS | – | 0.056 ± 0.002 | – | – |
| GAL-GAL-CONS4 | – | 0.22 ± 0.03 | – | – |

the control of *GAL1p* and *GAL10p* promoters along with Gal3p-Syn4.1 downstream of *GAL3p* promoter to create a single feedback strain, XYL-REG[SF]. In this strain, the plasmid carrying Gal3p-Syn4.1 downstream of *GAL1p* promoter necessary for dual feedback was excluded. We also tested the effect of constitutively expressing Gal3p-Syn4.1 by placing the gene downstream of *TEF1p* promoter (XYL-REG[C]). Both the strains have a growth rate of 0.12 per hour and a final OD of ~8, a 20% decrease in growth rate, and a 27% decrease in final biomass compared to the dual feedback design (Supplementary Fig. 8B). This clearly showcases the growth benefits of the wild-type-like dual-positive feedback system in XYL-REG.

**XYL regulon enables better growth on native sugars**. Since mutations on Gal3p-Syn4.1 are not in the sugar-binding pocket, it retains its galactose-binding function (Fig. 3d). To test if Gal3p-Syn4.1 can still function as a galactose transcriptional regulator, we transformed the plasmid containing Gal3p-Syn4.1 in a *GAL3Δ* strain to create GAL-Gal3p-Syn4.1. The strain and the unmodified parent strain, W303-1a, when grown in SC medium containing 2% galactose showed similar growth rates of 0.3 per hour and reached similar cell densities (Supplementary Fig. 9A). Finally, since strains containing the regulon switch ON gene expression only when xylose is present, we hypothesized that XYL-REG would have better growth fitness than XYL-CONS under conditions where the regulon is either uninduced or repressed. We tested growth of these strains in sucrose and glucose, to assess if regulated expression of xylose catabolic genes provides improved fitness even under non-inducing conditions, compared to constitutive expression. As expected, the growth rate of XYL-REG was higher than that of XYL-CONS in both sugars. The strains having the regulon grew at ~0.3 per hour while the XYL-CONS had a growth rate of 0.24 per hour in both sugars (Supplementary Fig. 9B, C). Comparing the growth of strains with and without regulon in multiple sugars, it is clear that presence of regulon upregulates necessary genes for sugar metabolism only when the sugar is detected. This induction system leads to high-growth rates and also prevents metabolic burden in the presence of non-inducing sugars (Table 1). Thus, a regulon-based strategy not only results in faster growth rates to higher final cell densities and more complete sugar consumption in the non-native sugar it is designed to respond to, but also exhibits lower metabolic burden and improved growth in alternate sugars.

**Transcriptomic analysis of strains**. To provide insight into the genes differentially expressed in REG strains (GAL-REG (WT) and XYL-REG) when compared to CONS strains (GAL-CONS and XYL-CONS) that result in vastly different growth phenotypes, we carried out RNA-seq to profile their transcriptome. We used triplicates of these strains grown in their respective carbon sources, harvested them during mid-exponential phase of growth and used them for RNA-seq. We found that the transcriptome profiles of the REG and CONS strains varied drastically

(Supplementary Fig. 10). Next, we carried out a differential gene expression analysis between XYL-REG and XYL-CONS as well as between GAL-REG (WT) and GAL-CONS strains using the limma and edgeR packages. Genes with a statistical *p*-value <0.05 after Benjamini–Hochberg correction were considered as differentially regulated. A total of 4202 genes were differentially regulated between GAL-REG and GAL-CONS strains and 3314 genes between XYL-REG and XYL-CONS strains (Fig. 6a, b). We reasoned that if there are genes either directly or indirectly regulated by Gal4p, they would be differentially expressed in not just the strains grown in galactose (GAL-REG vs. GAL-CONS) but also in strains grown in xylose (XYL-REG vs. XYL-CONS). Further, we also hypothesized that since both the CONS strains lack regulation for sugar detection, those strains should exhibit a starvation-like response. To test both, we decided to select for genes that are up- and downregulated in both the differential gene expression analyses. We found 452 genes that were upregulated and 507 genes that were downregulated genes in both the regulon strains (Fig. 6c).

Next, we evaluated the functional relation between these genes by examining the Gene Ontology (GO) biological process terms that are enriched in up- and downregulated genes. Of the genes that are upregulated in the REG strains, we found 36 enriched GO terms, including those relating to mitochondrial translation and transport, cell division, ATP production, protein import, etc. We also found 11 GO biological process terms in the case of genes that were downregulated, which were involved in processes such as fatty acid and lipid metabolism, DNA repair, response to stimulus, etc. (Fig. 6d and Supplementary Fig. 11). Out of the genes enriched under the GO term for response to stimulus, 58 genes were associated with stress response. Since, we hypothesized many of these genes are regulated by Gal4p, we extracted genes that are known to be regulated by Gal4p from YEASTRACT[41] and compared them with differentially regulated genes from our analysis. We identified 181 genes as hits from the overlap of the two sets (Fig. 6e and Supplementary Data 1). GO term analysis of the genes show that many of the processes previously enriched in upregulated genes such as cell division, cell wall organization, mitochondrial translation, and membrane transport were enriched, suggesting that upregulation of these vital cellular pathways are a direct consequence of activating the GAL regulon. Interestingly, none of the previously enriched GO process terms from the downregulated genes were enriched in this analysis indicating that GAL regulon is probably not involved in downregulation of genes observed in REG strains. Between the REG and CONS strains, we also analyzed for differentially expressed genes that code for transcription factors (TFs) (Fig. 6f and Supplementary Data 2). We identified 29 TFs that were upregulated in the CONS strains, of which several are involved in response to stress, nutrient starvation, and DNA replication stress (e.g., *CBF1*[42], *RPH1*[43,44], *GSM1*[45], *HAA1*[42], *WAR1*[46], *YOX1*[42,47], *SUT2*[48], *MIG3*[49,50], and *GCN4*[51]). TFs involved in gluconeogenesis and glyoxylate cycle (*RDS2*[52]) as well as drug resistance (*RDR1*[53] and *STB5*[54]) were also upregulated in CONS strains.

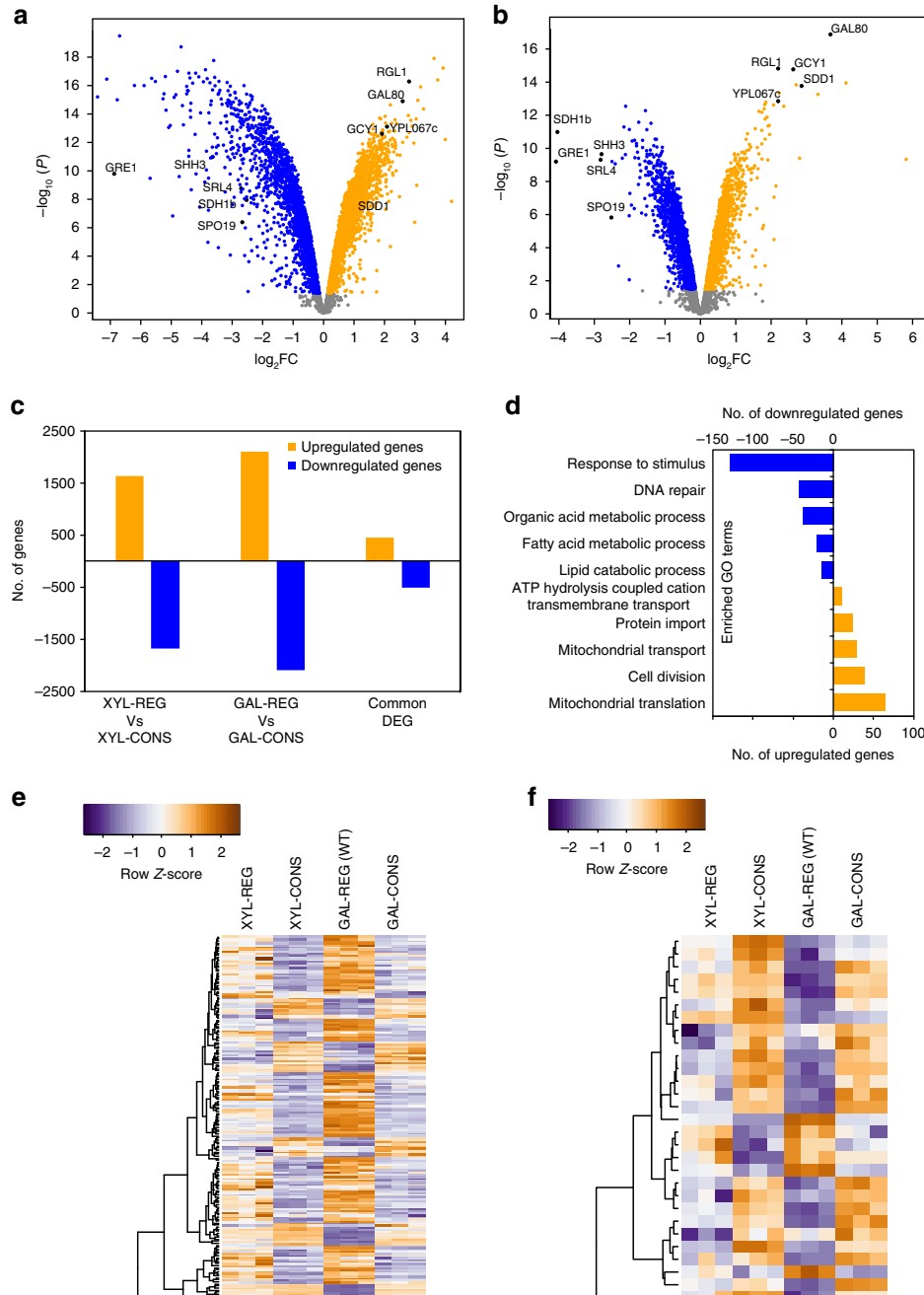

**Fig. 6** RNA-seq analysis of REG and CONS strains. Differentially expressed genes (DEGs) between **a** GAL-REG (WT) vs. GAL-CONS grown on galactose and **b** XYL-REG vs. XYL-CONS grown on xylose. DEGs that are upregulated are shown in orange while downregulated genes are shown in blue. Genes that are not differentially regulated are shown in gray. DEGs common to both GAL and XYL sets with high-fold change values are labeled and shown in black. **c** Number of genes that are differentially expressed between REG and CONS strains as well as genes common between the two pairs. **d** Relevant GO biological process terms of common DEGs. Negative numbers represent downregulated genes and positive ones represent upregulated genes in the REG strains. Heatmap of normalized log counts of DEGs (**e**) controlled by Gal4p, and **f** that are transcription factors (TFs). Identities of all genes in the heatmaps are provided in Supplementary Data 1 and Data 2

Upregulation of the above-mentioned TFs along with GO term enrichment of processes involved in DNA repair as well as upregulation of several stress response genes in the CONS strains reaffirms our hypothesis that CONS strains exhibit a starvation- and stress-like response when grown in carbon sources for which nutrient sensing systems are absent. In the case of REG strains, TFs responsible for cell wall production (INO4[55,56]), cell cycle progression (SWI5[57,58] and HCM1[59,60]), and flocculation suppression (SFL1[61]) were upregulated (Fig. 6e). Taken together with growth studies, the data suggests GAL regulon-controlled upregulation of pathways involved in growth such as cell wall maintenance, cell division, mitochondrial biogenesis, and cell cycle progression support the observed growth phenotype. On the other hand, unregulated constitutive expression of sugar metabolizing genes seems to trigger stress, starvation, and DNA damage responses.

## Discussion

In this study, we compared regulon-assisted control that is prevalent in nature to constitutive expression strategy, which is widely used in synthetic biology and metabolic engineering communities for non-native sugar assimilation. By first assessing the growth of *S. cerevisiae* on a native sugar, galactose, using the two strategies, we provide evidence that GAL regulon offers significant advantages for promoting growth compared to constitutive expression. We attribute this to the fact that apart from dynamically regulating the (upstream) Leloir pathway genes, the GAL regulon also regulates hundreds of other downstream metabolic and regulatory genes[27,29]. We show that activation of these pathways act synergistically and provide growth benefits to yeast on galactose. With the knowledge that GAL regulon can enhance growth and that the downstream genes were not galactose metabolism-specific, we adapted it for heterologous xylose metabolism in *S. cerevisiae* and circumvented the need for extensive genome-scale engineering that would otherwise be needed for synthetic regulon construction. We engineered the GAL regulon into a xylose (XYL) regulon and with minimal metabolic engineering, obtained better growth and final cell density compared to constitutive expression of upstream xylose metabolic genes. It should also be noted that this is in stark contrast to most published studies where growth rate in xylose prior to adaptive evolution is low. The growth rate of XYL-CONS, 0.06 per hour is obtained due to superior *XYLA*3*, which was engineered by Lee et al.[39], who report similar growth rate in their work[39]. While a number of studies have obtained higher growth rates than what we report for XYL-REG, those strains have all been engineered extensively with overexpression of all of the non-oxidative pentose pathways as well as adaptive evolution[11–13,62]. We would also like to point out that mRNA expression levels of *XYLA*3* and *XKS1* from XYL-REG strain is several folds higher than expression in XYL-CONS strain and can, to some extent, contribute to the observed increase in growth rate (Supplementary Note 5 and Supplementary Fig. 7). But, transcriptomic analysis by RNA-seq revealed that genes responsible for cell wall biogenesis, mitochondrial biogenesis, and ATP biosynthesis were upregulated in the REG strains, suggesting that GAL regulon-mediated activation of downstream genes also plays a major role in promoting fast growth of the REG strains. On the other hand, genes involved in response to stress, starvation, DNA damage, and lipid metabolism were upregulated in the CONS strains as a consequence of being forced to metabolize unrecognized nutrients. Thus, the GAL regulon seems to aid in growth by upregulating several growth-related pathways and transcription factors while suppressing stress and starvation responses, which are upregulated in strains with unregulated nutrient catabolic pathways. While metabolite sensing is increasingly employed in metabolic engineering, they are used to *trans*-activate only a small set of genes[3,4,63,64]. Other approaches such as transcription factor-based engineering, which involves deletion or overexpression of specific transcription factors have also been carried out for non-native sugar metabolism such as xylose or cellobiose[65,66]. However, as far as we know, this is the first known engineering effort that rationally couples nutrient sensing to direct global cellular state for fast-growth while also repressing stress and starvation responses that is generally observed when *S. cerevisiae* is grown in a non-native sugar[66,67]. We also demonstrate that this cellular state is congruent with observed transcriptional and phenotypic responses on native and non-native nutrients.

Maintaining tight regulatory control along with rapid and robust response to a nutrient is an essential characteristic for nutrient-induced regulon activation. For growth fitness in a competitive environment, yeast evolved mechanisms for initiating rapid and robust response to assimilate available galactose through dual feedback loops while maintaining tight regulatory control. In this work, we show that a single protein (Gal3p-Syn4.1) involved in repressor (Gal80p) sequestration when driven by promoters of different expression strengths can create a dual-positive feedback loop that has increased response to the inducer, exhibits bistability, and is resistant to noise. Further, this dual-positive feedback regulatory structure supports higher growth rate and cell density when compared to single-positive feedback or constitutive GAL activation systems. The dual-positive feedback loop conceived in this study is a synthetic implementation of the native GAL system. Thus, the high-growth rate obtained by the XYL-REG strain is a synergistic effect of engineering the native regulatory and metabolic architecture of the GAL regulon at various levels such as sensing, transduction, and metabolism. Altogether, this work provides a paradigm of engineering semi-synthetic regulons for nutrient assimilation and highlights not only the importance of sensing nutrients, but also how they integrate into other cellular functions to ensure activation of growth responses and repression of starvation/stress responses. This approach can be easily extended for other abundant, but non-native, nutrients. We suggest that the regulon engineering strategy is a rational and potentially faster, and possibly more elegant approach than prevailing strategies.

## Methods

**Strains and plasmids**. All the list of plasmids and strains used are listed in Supplementary Table 2 and Supplementary Table 3, respectively.

**Materials**. Strain W303-1a (*MATa {leu2-3,112 trp1-1 can1-100 ura3-1 ade2-1 his3-11, 15}*), and plasmids pkT209 and pBK415 were obtained from Euroscarf (Frankfrut, Germany). All enzymes for cloning were purchased from NEB (Beverly, MA). DNA primers were ordered from Operon Inc. (Huntsville, AL). Sequencing of plasmid DNA was outsourced to Operon Inc. All primer sequences are listed in Supplementary Table 4. Growth media and chemicals were purchased from Amresco (Solon, OH) or RPI Corp (Mount Prospect, IL). Ampicillin was obtained from RPI and G418 sulfate from Life Technologies (Grand Island, NY). 5-Fluoroorotic Acid (FOA) was purchased from Zymo Research (Irvine, CA). E.Z.N.A.® Plasmid Mini Kit I, PCR Purification and Gel Extraction Kits were obtained from Omegabiotek (Norcross, GA). *XYLA*3* DNA sequence was provided by Prof. Hal S. Alper (University of Texas at Austin) and was synthesized from Twist Biosciences (San Francisco, CA). *GAL2-2.1* DNA sequence was provided by Prof. Bernard Hauer (University of Stuttgart, Stuttgart, Germany). Complete Supplement mixture without Histidine (His), Leucine (Leu), Uracil (Ura) and Tryptophan (Trp) mixture was obtained from Sunrise Science Products, Inc (San Diego, CA).

**Strain construction**. The yeast strain W303-1a was used for constructing all the strains used in the study. The knock out protocol "Delitto perfetto"[68] was used with modifications. Instead of using CORE (Counterselectable Reporter) cassette containing *URA3* and *KANMX* markers, only *URA3* was used. Selection was performed in Synthetic Complete (SC) medium without uracil and counterselection using SC medium with 1 g/l of 5-FOA. *URA3* gene was PCR amplified from pkT209 plasmid using primers with 40 bp flanking ends, which are homologous to the gene ends to be knocked out. The deletion cassette was inserted through homologous recombination by lithium acetate transformation protocol of Gietz[69]. The transformed cells were selected for *URA3* cassette insertion by selecting in SC-Ura medium and confirmed using colony PCR. To remove the cassette, two colony PCRs were performed to amplify the flanking ends of gene using primers with overlapping ends. The two fragments were spliced using Overlap-Extension PCR (OE-PCR), transformed and selected in SC + FOA plates and confirmed using colony PCR. *GAL2*, along with its native promoter, was amplified from yeast genome and single point mutations at different sites were introduced through OE-PCR. *GAL2p-GAL2-2.1-TEF1t* construct with *Xho*I and *Not*I restriction sites were restriction digested and ligated with pRS405 to create pRS405-*GAL2p-GAL2-2.1-TEF1t*. The plasmid was linearized by making a single cut with *Eco*RI at the *LEU* locus, transformed in VEG16 strain and selected for colonies in SC-Leu medium supplemented with 2% glucose and confirmed through colony PCR. The *Promoter-GFP-TEFt* constructs from pRS426 was restriction digested with *Bam*HI and *Sal*I and cloned into pRS406. The integrative vectors were then linearized with *Nde*I and used to transform VEG16 strain.

**Plasmid construction**. The plasmid for screening and selection, pVEG8 was built through two sequential cloning steps. The bidirectional promoters *GAL1p/GAL10p* and *HXT7t* terminator were amplified from the yeast genome. *KANMX* gene was amplified from plasmid pBK415; *EGFP*, and *ADH1t* terminator were amplified

from pkT209. All of them were spliced using OE-PCR, restriction digested with *Bam*HI and *Sal*I, ligated and cloned into pRS426 to create pVEG7. *GAL3* gene along with its native promoter was amplified from the yeast genome, spliced with *TEF1t* terminator amplified from plasmid pkT209 and cloned into pVEG7 backbone using *Bam*HI and *Not*I restriction sites to create pVEG8. To create pCONS-GAL, three constitutive promoters, *TEF1p*, *TPI1p*, and *GPM1p*, three terminators, *TEF1t*, *ADH1t*, and *HXT7t* along with genes *GAL1*, *GAL7*, and *GAL10* were amplified from the genome of *S. cerevisiae* using colony PCR with primers containing appropriate flanking regions for OE-PCR. Three constructs *TEF1p-GAL1-Hxt7t*, *TPI1p-GAL10-ADH1t*, and *GPM1p-GAL7-TEF1t* were built using OE-PCR and assembled onto pRS426 backbone using DNA assembler[70]. Two plasmids for xylose metabolism were built using genes *XYLA*3* from *Piromyces sp.* (codon optimized), and *XKS1*, *TAL1*, and *GAL2-2.1* from *S. cerevisiae*. KANMX and EGFP were replaced by *XYLA*3* and *XKS1* in pVEG7 to create pVEG11. Similarly, *TAL1* and *GAL2-2.1* were cloned into the promoter-gene-terminator construct of pVEG7, but in pRS423 backbone to create pVEG12. The *GAL1p/GAL10p* promoters of pVEG11 and pVEG12 were replaced by divergent *TEF1p* and *TPI1p* promoters to create pVEG10 and pVEG13, respectively. The Syn4.1 mutant of *GAL3* was subcloned into pRS414 from pVEG8mut to create pVEG16*. Finally, Syn4.1 mutant of *GAL3* was also placed under *GAL1p* promoter and cloned into pRS415 backbone to create pVEG17*. Promoters of genes *GAL1*, *GAL3*, *GAL80*, *GAL7*, *GAL10*, *FUR4*, *TEF1*, *TPI1*, *GPM1*, *PCL10*, *REE1*, *LAP3*, and *MTH1* were amplified from yeast genome, spliced with *EGFP-ADH1t* construct using appropriate primers from pVEG8, cloned onto pRS426 backbone to create the respective *pRS426-Promoter-EGFP-ADH1t* constructs.

**Antibiotic selection and fluorescence screening**. *S. cerevisiae* strain, VEG16 or VEG20 were transformed with mutant libraries of pVEG8 using established protocols[69] and recovered for 6 h in 1.2 ml of YP supplemented with 2% of sucrose and xylose before plating on the agar plates with same medium, supplemented with 100 μg/ml of G418 sulfate. The plates were incubated at 30 °C for 2–3 days and colonies were streaked in SC-Ura medium with glucose. Grown colonies were then inoculated in both 2% sucrose and 2% sucrose supplemented with xylose (2% or 0.2% as mentioned) in 96-well plates and incubated for 18 h in a shaker. Fluorescence (excitation at 488 nm and emission at 525 nm) and OD$_{600}$ were measured in a Spectramax M3 spectrophotometer to obtain RFU/OD$_{600}$. Only strains that exhibited low-basal fluorescence in sucrose and higher fluorescence than wild-type in xylose were taken for further screening. Characterization of the fluorescence profile of mutants were carried out by inoculating the strains on medium with SC-Ura with sucrose supplemented with different concentrations of xylose or galactose and RFU/OD$_{600}$ was measured after 18 h of incubation in a plate shaker.

**Dose-response curve for fluorescence**. For all dose-response curve experiments, the cells were grown in 2% sucrose unless specified. The cells were first pre-grown for 24 h, except for hysteresis experiments where the cells were grown in media supplemented with 4% xylose for full induction. The cells were diluted 100-fold in SC medium containing sucrose and specified concentrations of galactose or xylose. They were incubated at 30 °C in a microplate shaker and RFU/OD$_{600}$ was measured in a spectrophotometer with excitation at 488 nm and emission at 509 nm. For comparing dual and single feedback cell populations, fluorescence was measured using Attune Nxt5 flow cytometer. Blue laser (488 nm) was used for excitation. At least 10,000 cells were measured for each of the flow cytometry experiments.

**Growth studies**. The strains were grown overnight in appropriate dropout SC medium supplemented with sucrose. They were washed thrice in the growth medium to be inoculated and then diluted to an initial OD$_{600}$ of 0.1 in the same medium with appropriate sugar (2%), incubated in 5 ml test tubes, and OD$_{600}$ was measured every couple of hours. Growth studies with xylose or galactose were carried out in 250 ml shake flasks containing 20 ml of media. Growth studies with sucrose or glucose were carried out in 15 ml test tubes containing 5 ml of media. For measuring concentration of extracellular xylose, samples collected during OD$_{600}$ measurement were centrifuged at 10,000 × *g* for 1 min and the supernatant was stored at −20 °C.

**Media and transformation**. Yeast strains were grown in YPA medium or SC medium (Yeast nitrogen base (1.67 g/l), ammonium sulfate (5 g/l), complete supplement mixture without His, Leu, Ura and Trp (0.6 g/l)) with appropriate nutrient. Luria Bertani (LB) broth and LB agar plates with 100 mg/l of ampicillin when required were used for all *E. coli* propagation and transformation experiments. *E. coli* NEB5α was used to transform the ligated mixture to create all the plasmids described using MES transformation except for mutant libraries, which were created by electroporating the ligation mixture. The plasmids were sequenced and transformed into the appropriate yeast strain using the protocol of Gietz[69].

**Mutagenesis**. Random mutagenesis libraries were created by error prone PCR with 0.3 ng/μl of template plasmid, 0.2 mM dATP, 0.2 mM dGTP, 1 mM dCTP, 1 mM dTTP, 5 mM MgCl₂, MnCl₂ (0.05 mM for mutagenesis on the entire protein and 0.3 mM for mutations on the loops), 0.05 U/ml *Taq* DNA polymerase, and 0.4

mM of the forward and reverse primers. The reaction was amplified using the following PCR cycle conditions: 95 °C denaturation, 5 min; 16 cycles of 95 °C denaturation, 30 s; 46 °C annealing, 45 s; and 68 °C extension, 3 min, followed by 68 °C extension for 5 min. For mutagenesis on the interaction loops, the cycle number was increased to 25. The mutated gene was spliced with *GAL3p* promoter and *TEF1t* terminator using OE-PCR, restriction digested, cloned into pVEG7 plasmid and electroporated to *E. coli* NEB5α cells. Five transformants were randomly chosen and their gene sequenced to determine the error rate of the library. For saturation mutagenesis, forward and reverse primers with NNK degenerate nucleotides at position 109 were used to create fragments that were spliced using OE-PCR, restriction digested with *Bam*HI and *Not*I, ligated, and cloned into pVEG7 background. For synthetic shuffling, five degenerate primers that code for either the wild-type or mutated nucleotide covering all eight mutations obtained from random mutagenesis were used to amplify fragments from *GAL3-2.1* variant, spliced with OE-PCR, restriction digested, and cloned into pVEG7 background.

**Extracellular xylose measurement**. Xylose concentration was measured using an Agilent HPLC system equipped with a Hi-Plex H-column and detected using 1260 Agilent ELSD detector. Mobile phase was 0.1% trifluoroacetic acid (TFA) with a flow rate of 0.6 ml/min. The ELSD detector's nebulizer and evaporation temperature were set at 30 °C and nitrogen flow rate at 1.6 SLM (standard liter per minute).

**RNA analysis**. Triplicates of strains WT (GAL-REG), XYL-REG, GAL-CONS, and XYL-CONS were grown in their respective carbon source (galactose or xylose) until mid-exponential phase and approximately 2 × 10⁷ cells were washed twice in water, pelleted and stored at −80 °C. RNA extraction as well as library preparation and sequencing were outsourced to Genewiz, Inc. (South Plainfield, NJ). RNA-seq was performed on Illumina HiSeq. Possible adapter sequences and nucleotides with poor quality were trimmed, sequence reads shorter than 50 bp were excluded and the remaining were aligned to the reference genome W303 obtained from Saccharomyces Genome Database (http://www.yeastgenome.org) along with *XYLA*3* sequence. The obtained gene count data was normalized based on library size, converted to cpm (counts per million) using edgeR package. To prevent skewing of data, genes that were either deleted or overexpressed were removed from the data set for further analysis. For differential gene expression analysis, limma package with voom transformation was applied to the samples. Finally, a linear model was fit to each gene using limma (lmfit function) and differential gene expression was analyzed. Genes with Benjamini–Hochberg corrected *p*-values <0.05 were considered as differentially expressed.

**Gene Ontology term enrichment analysis**. GO term enrichment analysis was carried out using the GO term tool from Princeton University (http://go.princeton.edu/) by querying for commonly up- or downregulated genes separately, after applying Bonferroni correction and with a cutoff *p*-value of 0.05. The obtained GO terms were summarized using REVIGO toolbox with default settings.

**Computational modeling**. The ODE model from Venturelli et al.[35] was used with minor modifications. Since the regulatory network remains the same, but the mode of activation is either galactose or xylose, we retained all the parameters used by the previous study. In our case, Gal3p-Syn4.1 induction by xylose is weaker than Gal3p-WT activation by galactose. Since the model does not differentiate free Gal3p with sugar bound Gal3p, we assume Gal3p-sugar-Gal80p interaction (kf83 and kr83) to be a lumped parameter that represents Gal3p-based Gal4p activation. In our model, Gal1p was not considered since it has been deleted in our strains and represents the single feedback model. For dual feedback, the expression of Gal3p under *GAL1p* promoter was modeled by adding a second Gal3p production term under *GAL1p* promoter. In both the models, EGFP expression under *GAL1p* promoter was modeled. Since EGFP has low degradation rate, decay of EGFP was ignored. Further details on parameters as well as ODE equations for both models are provided in the Supplementary Note 3 and Supplementary Table 5.

**Statistics and data reproducibility**. All the experiments were conducted using biological triplicates that were carried out on three different days to calculate measure of variability between the samples. All the data shown are mean with error bars representing standard deviation.

**Data availability**. The RNA-seq data in this study have been deposited in the National Center for Biotechnology Information (NCBI) Gene Expression Omnibus (GEO) with accession code https://www.ncbi.nlm.nih.gov/geo/query/acc.cgi?acc=GSE110818. All other data supporting the findings of this study, including computer codes are available from the corresponding author upon reasonable request.

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

## Acknowledgements

This material is based upon work supported by National Science Foundation (NSF) Grant No. CCF-1421972, Eunice Kennedy Shriver National Institute of Child Health and Human Development (NICHD) of the National Institutes of Health (NIH) under award number DP2HD091798, Tufts University Tufts Collaborates! grant to N.U.N. and The Shirley and Stanley Charm Scholarship in Food and Biotechnology, Tufts University to V.E.G. The article processing charge was funded by Tufts University Faculty Research Awards Committee (FRAC) for Open Access Publication and Graduate Open Access Funding. We thank Dr. Hal S. Alper (University of Texas at Austin, Austin, Texas) for providing the sequence for *XYLA*3* and Dr. Bernhard Hauer (Institute of Technical Biochemistry, University of Stuttgart, Stuttgart, Germany) for providing the sequence for *GAL2-2.1*. We also would like to thank Todd C. Chappell for valuable discussions and Josef R. Bober for help with growth experiments.

## Author contributions

V.E.G. and N.U.N. planned the experiments, wrote the manuscript, and analyzed the results. V.E.G conducted all the experiments.

## Additional information

**Competing interests:** The authors declare no competing interests.

