## [Peer Review File(PDF 567 kb) · Nature Communications]

Reviewers' comments:

Reviewer #1 (Remarks to the Author):

In this study, Gopinarayana and Nair engineer the galactose-responsive pathway from *Saccharomyces cerevisiae* to use both galactose (which it uses natively) and xylose (non-native). To do this, they use *S. cerevisiae*'s galactose regulation machinery and modify it using directed evolution of Gal3p to enable it to respond to xylose in addition galactose. In addition, they altered the native dual positive feedback signal transduction system in their xylose synthetic design, comparing alternative forms of the feedback network. The designs are compared with strong constitutive expression of the genes to weigh the benefits of the more complex regulon-based integration. They find that their synthetic system is able to use xylose and exhibits higher growth rates and cell densities than simpler synthetic systems based on constitutive expression.

Although the manuscript and results appear to be technically reasonable, with an exception I note below, I generally found the study to lack broad relevance and impact. My rationale for this is based on three points:

1. There are a reasonable number of previously published studies that aim to engineer feedback control to improve biosynthetic pathways. Although constitutive or inducible promoters are common in metabolic engineering, there is an increasing recognition of the importance of designing dynamic control systems, so although I find the work to be interesting I do not find it to be particularly novel. A couple examples of work in this area include Dahl, et al. – Nature Biotech 2013 (PMID: 24142050) and Zhang, et al. – Nature Biotech 2012 (22446695).
2. The constitutive promoter designs they compare their synthetic system against are a bit of a straw man. It is likely possible to carefully tune expression of even constitutive/inducible promoters to optimize for xylose and galactose utilization. By simply putting genes under control of a strong constitutive promoter with no further optimizing it is not surprising that results are poor. To be clear, the feedback network likely offers advantages, but pitting the synthetic system against a design with no further optimization is not necessarily a fair comparison. In addition, it may be the case that the benefits of the feedback structure are only apparent in dynamic situations with variability and uncertainty in nutrient levels.
3. It is not clear that the approach is generalizable. It is not clear why one vs. two feedback loops are compared, as opposed to a different set of feedback loops or some other choice, for instance. As a result, it is difficult to see how insight from this study will transfer to other projects beyond those directly involving GAL.

On the technical side: in many cases replicates are based on two measurements. These experiments should be conducted at least in triplicate.

Reviewer #2 (Remarks to the Author):

This work introduces a novel strategy to engineer *Saccharomyces cerevisiae* to utilize a non-natural carbon source, xylose, by harnessing the natural galactose regulon in this organism. This is a very interesting study that includes some fundamental research on galactose metabolism, protein engineering, and metabolic engineering, and should be of interest to a wide range of researchers. The study claims that using the GAL regulon to express the catabolic enzymes for xylose utilization has some advantages over constitutive expression of the catabolic pathway. This idea has a lot of potential. However, this study has some serious problems and unanswered questions that need to be addressed:

- 1) The authors should be cautious with their wording, specifically about calling this the "first design rules for incorporating a non- native nutrient assimilation system in a synthetic biological

system". What they call "design rules" are the well-known principles of the galactose inducible system in yeast, and are not new. There are also several examples of previous yeast metabolic engineering studies in which close attention was paid to the design of systems to utilize and co-utilize natural and non-natural substrates, not from the standpoint of catabolic routes, but rather strategies for substrate assimilation (xylose, arabinose, glucose, cellobiose, acetate -- see studies by Jamie Cate, Yong-Su Jin, and others). This study may represent a novel strategy in this theme, but it is an overstatement to call it "the first"

2) The text contains several statements about previous studies that could be interpreted as somewhat derogatory. For example calling them or implying that they are simplistic, disregarding, or inelegant.

3) When they compare the regulon and constitutive systems in both, galactose (natural GAL regulon) or xylose experiments, they do not address the possibility that the effects they see are caused (at least in part) by differences in promoters strength. On average, GAL1p, GAL10p, and GAL7p (in galactose) are stronger promoters than TEF1p, GPM1p, or TPI1p – it is not clear how strong these inducible and constitutive promoters are in xylose. Furthermore, the strength of GPM1p, and TPI1p promoters has only been characterized in glucose, and they may be weaker in galactose or xylose. The difference in strength between these promoters could explain at least some of the positive effect they see in utilizing the GAL system to drive galactose and xylose metabolism. Also, the claim that they use the strongest-known constitutive promoters is not true. The TDH3p (also known as GPD1p) promoter is stronger than TEF1p, GPM1p, or TPI1p, at least in glucose. Again, the strength of these promoters should be assessed in the carbon sources in which they are utilized (xylose or galactose), as they may have different strengths in different carbon sources. They need to rule out that differences in promoter strength plays a significant role in their results.

4) It is not clear why the downstream GAL genes should benefit xylose catabolism. Galactose involves the Leloir pathway while xylose involves the pentose phosphate pathway. The authors mention in the discussion that these downstream genes are not galactose-metabolism specific, but they provide no data or references for this claim. The experiments done in this study do not rule out the possibility that the differences in promoter strengths are responsible for the REG systems being superior to the constitutive promoters. They also do not offer any mechanistic explanation for how these downstream genes could enhance xylose catabolism (not even hypothetical). If downstream genes are so important for the efficient utilization specifically of galactose, as opposed to say glucose, or maltose, why should those same downstream genes benefit xylose assimilation? Conversely, if they are not specific for galactose, why are they so tightly regulated by the GAL system if they could also enhance glucose or maltose catabolism?

One way to obtain evidence that the downstream genes enhance xylose catabolism may be to induce a XYL-CONS strain growing in xylose with low concentrations of galactose (in a strain in which low levels of galactose can induce the downstream genes). If low levels of galactose can enhance the strain's growth on xylose while demonstrably inducing the downstream GAL genes, it would provide more convincing evidence to their claim that these downstream genes benefit xylose metabolism.

5) In lines 137 and 138, it is not clear how saturation mutagenesis of one position (109) can lead to 3000 variants. Please clarify.

6) They need to explain how they measured or estimated the diversity of their error prone PCR libraries.

7) What are the implications of getting two distinct populations in their flow cytometry measurements as they increase xylose concentration? It seems that the genes are either completely on or off, with no intermediate level of expression. This is reported but not discussed.

They must also summarize the similarity of their results with the observations of Bradman et al, rather than just mentioning consistency and referencing their study.

8) The comparison of their strains to previously developed xylose-utilizing strains is a bit misleading. In their discussion, the authors compare the growth rates of their strains in complex YP medium with the growth rates of previous strains (Zhou et al, and Lee et al) obtained in SD medium. If they make the more fair comparison using the growth rate of their strains in the same medium as previous measurements (SD medium), then their XYL_CONS and XYL-REG strains are both substantially slower than other constitutive strains. This raises the question of why their constitutive strain is so slow, compared to previously developed constitutive strains? It also suggests that the XYL-REG strategy is not effective at achieving the growth rates obtained in existing (constitutive) xylose-utilizing strains.

(This again raises the possibility that their results are largely due to better expression of catabolic enzymes from the strong GAL promoters relative to the constitutive promoters they used, which may be a lot weaker than the authors assume – especially in galactose or xylose).

9) Most of their experiments were done using 2-micron plasmids, including when developing the XYL-REG and XYL-CONS strains, and measuring the expression of other downstream genes under galactose or xylose. However, it is well known that 2-micron plasmid can produce transformants with widely heterogeneous phenotypes (not all colonies are equal). They must report how homogeneous were the phenotypes they observed across transformants, or report how many colonies they needed to screen to find the strains they analyzed in more depth.

10) The authors overstate the impact of their findings when they claim to have found an “outline for the basic design guidelines for engineering synthetic non-native regulons for nutrient assimilation”. This study is indeed very interesting, and has great potential, but it is difficult to find any general rule here. The system may have worked (provided they rule out other explanations above) because of structural similarities between xylose and galactose, the availability of the natural galactose regulon in yeast, and ultimately Gal3p’s natural ability to recognize xylose. But how are the rules they claim to have identified any different from what we already know about the GAL regulon? Also, if these rules are to be used as guidelines, how can they be extrapolated for other non-natural substrates, say formate, methanol, or cellobiose, for which GAL3 is less likely to be an effective substrate sensor?

11) Figures 4G and S5 and the experiments they describe are not very clear. What values correspond to cells grown in sucrose (or glucose)? They do not seem to be in the figure. If they are not shown, then what is the point of the experiment? Is it not to compare levels of induction of downstream genes by xylose or galactose with respect to a baseline sugar (sucrose or glucose)?

12) They need to clearly define all the variables and constants in their models, as well as list all their assumptions (some assumptions are mentioned in the methods section, but it is not clear that this is a complete list).

13) Color code is missing for Fig S2.

14) There is an author’s comment left in Fig S6

15) Typo in line 230: “include” is written twice

Reviewer #1 (Remarks to the Author):

In this study, Gopinayanan and Nair engineer the galactose-responsive pathway from *Saccharomyces cerevisiae* to use both galactose (which it uses natively) and xylose (non-native). To do this, they use *S. cerevisiae*'s galactose regulation machinery and modify it using directed evolution of Gal3p to enable it to respond to xylose in addition galactose. In addition, they altered the native dual positive feedback signal transduction system in their xylose synthetic design, comparing alternative forms of the feedback network. The designs are compared with strong constitutive expression of the genes to weigh the benefits of the more complex regulon-based integration. They find that their synthetic system is able to use xylose and exhibits higher growth rates and cell densities than simpler synthetic systems based on constitutive expression.

Although the manuscript and results appear to be technically reasonable, with an exception I note below, I generally found the study to lack broad relevance and impact. My rationale for this is based on three points:

1. There are a reasonable number of previously published studies that aim to engineer feedback control to improve biosynthetic pathways. Although constitutive or inducible promoters are common in metabolic engineering, there is an increasing recognition of the importance of designing dynamic control systems, so although I find the work to be interesting I do not find it to be particularly novel. A couple examples of work in this area include Dahl, et al. – Nature Biotech 2013 (PMID: 24142050) and Zhang, et al. – Nature Biotech 2012 (22446695).

We agree with the reviewer that there have been several previously published studies that use dynamic feedback control to improve production of valuable biochemicals and these studies have, in fact, shown the importance and advantages of having dynamic control over static control. These seminal works demonstrate the importance of how to deal with *native* signals (eg: stress response, oleic acid) that are *automatically* activated in response to stresses/metabolites of biosynthetic pathways/intermediates. In our work, we demonstrate how to *synthetically* activate global cellular response to a *non-native* signal and use it for the non-native nutrient catabolism. This type of engineering has not been done for any synthetic system, and thus, we posit that our work is unique and distinct from these and other seminal works. Further, dynamic control switches for metabolite sensing in yeast have mostly used fluorescence as their output with a few exceptions, such as malonyl-CoA sensing to control 3-HP production by David et al., 2016. While previous studies have focused on xylose sensing using bacterial transcriptional repressors with fluorescence as output in yeast, they have not extended it towards changing the genome-wide transcriptome. The novelty of our work is that we use a native regulatory system to sense a non-native nutrient and in response, modulate expression of hundreds of genes for xylose catabolism and growth. The addition of RNA-seq data further demonstrate how drastic the phenotypic changes are between the conventional design and our regulon design not just for a non-native nutrient xylose, but also for the native sugar galactose.

Reference:

David, F., Nielsen, J., & Siewers, V. (2016). Flux Control at the Malonyl-CoA Node through Hierarchical Dynamic Pathway Regulation in *Saccharomyces cerevisiae*. *ACS Synthetic Biology*, 5, 224–233. <https://doi.org/10.1021/acssynbio.5b00161>

The corresponding changes to the manuscript are included here:

In recent years, dynamic pathway regulation has yielded significant improvements in product titers using either native^{1,2} or heterologous transcription factors³⁻⁶.

2. The constitutive promoter designs they compare their synthetic system against are a bit of a straw man.

It is likely possible to carefully tune expression of even constitutive/inducible promoters to optimize for xylose and galactose utilization. By simply putting genes under control of a strong constitutive promoter with no further optimizing it is not surprising that results are poor. To be clear, the feedback network likely offers advantages, but pitting the synthetic system against a design with no further optimization is not necessarily a fair comparison. In addition, it may be the case that the benefits of the feedback structure are only apparent in dynamic situations with variability and uncertainty in nutrient levels.

We agree that not fine-tuning promoter expression could be an issue. However, we believe that the comparison between both strains is a fair one since neither the constitutive promoter system nor the synthetic regulon promoters has been optimized or tuned to balance expression of *XYLA*3*, *XKS1*, *GAL2-2.1*, and *TALI*. It is likely that further optimization of gene expression could lead to enhanced growth rate of *both* strains. We do not have any reason to believe that absolute expression-level in one system is less or more optimal than the other or that more significant improvements in growth rates could be achieved for either strain with further pathway optimization. In the case of galactose utilization, to rule out the possibility that constitutive promoters are decreasing growth rate and not the absence of the regulon, we have used a control strain GAL-CONS-*GAL4*, in which the initial catabolic enzymes are expressed constitutively while the regulon is still intact. In this strain, we were able to rescue a significant portion of the growth rate, thereby validating our claim that growth improvements are due to GAL4-mediated activation of the regulon (and downstream metabolic genes) rather than optimal expression of (upstream) Leloir genes.

We compare our feedback regulon system to constitutive expression system because the latter is the state of the art for catabolic pathway implementation. There has been no explicit demonstration that a feedback loop or dynamic regulation provides any benefit for improved growth on non-native nutrients. In fact, our data demonstrates that just the presence of feedback or dynamic pathway regulation at a single pathway level confers only minor benefits for growth (see growth rate of GAL-REG versus GAL-CONS-*GAL4* in Fig 1). This reasoning for comparing constitutive expression with a regulon is, in fact, part of our argument that regulatory control of downstream pathways should be considered an important aspect of catabolic pathway design/implementation. However, most current work in implementing catabolic pathways, do not consider this. Thus, this work is the first systematic demonstration that integration of synthetic nutrient assimilation within the host context is an important design criterion that has not been considered previously. We have also provided additional data from RNA-seq (Results section and Fig. 6) to demonstrate which genes are being directly activated under the REG strains (GAL-REG (WT) and XYL-REG) but not in CONS strains (GAL-CONS and XYL-CONS). Many of the genes code for pathways that are involved in cell division, mitochondrial biogenesis, ATP biosynthesis etc., which are important for cell growth and doubling.

We agree with the reviewer that feedback structure offers benefits that are largely apparent in dynamic situations. However, we also demonstrate (and now with additional RNA-seq data) that the primary benefit of the XYL and GAL regulons are provided by integration of the sugar-sensing into the native cellular metabolism and GAL4-based activation of many native genes that are required for high growth rates. As you point out here, and we concluded based on comparing WT and CONS-*GAL-GAL4*, when grown solely on galactose, the benefits of the feedback system are minor. The major benefit is due to GAL4-based activation of downstream genes.

The corresponding changes to the manuscript are included here:

The decrease in growth rate can either be attributed to inability of the strain GAL-CONS to activate the downstream genes of GAL regulon that are required for growth, or the difference in promoter strengths between the GAL and constitutive promoters that transcribe the Leloir pathway genes, or both. To determine true cause of growth rate decrease, we re-introduced Gal4p in GAL-CONS but deleted Leloir

pathway genes under its native promoter as well as *GAL3* and *GRE3* (which encodes for non-specific aldose reductase) to create the strain GAL-CONS-*GAL4* (**Fig 1A**). This re-factored, partially-coupled, system should enable activation of downstream genes through Gal1p-Gal80p-Gal4p pathway², but keeps the Leloir pathway genes out of the GAL regulon control. Thus, if the downstream genes of the GAL regulon assist in growth, the partially coupled strain should have growth rates higher than the GAL-CONS strain. On the other hand, if the observed decrease is a result of the difference in promoter strengths between native GAL promoters and constitutive promoters, the GAL-CONS-*GAL4* strain should have the same growth rate as that of the GAL-CONS strain. We tested the GAL-CONS-*GAL4* strain for growth in galactose, and observed that the strain recovers a significant portion of its growth fitness relative to GAL-CONS (**Fig 1B**) suggesting that the downstream genes under the control of the regulon *trans*-activated by Gal4p positively affect the ability of yeast to grow on galactose. It should be noted that constitutive promoters (expressed on multicopy plasmids) have higher expression strength than GAL promoters (**Fig S1 & supplemental note 1**), thereby suggesting the promoter strengths do not play a major role in dictating the growth rate of the strains tested. This can be seen when comparing growth of wildtype and GAL-CONS-*GAL4* strains (**Fig 1B**). Here, the benefits of using GAL promoters is demonstrated by the slightly higher growth rate. However, most of growth benefit is gained by activation of downstream (non-Leloir) genes.

3. It is not clear that the approach is generalizable. It is not clear why one vs. two feedback loops are compared, as opposed to a different set of feedback loops or some other choice, for instance. As a result, it is difficult to see how insight from this study will transfer to other projects beyond those directly involving GAL.

On the technical side: in many cases replicates are based on two measurements. These experiments should be conducted at least in triplicate.

Thank you for these insightful comments. The purpose of comparing the single and dual feedback loop was to demonstrate that a XYL regulon can be engineered to very closely resemble the GAL regulon and reap the benefits provided by that native system, which enables very high growth rates. It also provides benefits in dynamic situations as you already noted previously. We argue that the downstream genes activated by the regulon are largely independent of the carbon source provided. For example, upregulation of genes involved in cell division, ATP biosynthesis etc. is beneficial for growth on any carbon source. Thus, we believe that if Gal3p can be engineered to be activated by any other non-native nutrient, this system will provide significant benefits. We have shown here that this regulon can enable faster growth on two distinct substrates (galactose, xylose), and therefore believe that it is generalizable to other nutrients as well.

We apologize for neglecting to conduct some experiments in triplicate. We have addressed that concern in the revised manuscript.

All figures in the main manuscript were performed in triplicates, the figures were modified accordingly and the details are included within captions of all figures.

Reviewer #2 (Remarks to the Author):

This work introduces a novel strategy to engineer *Saccharomyces cerevisiae* to utilize a non-natural carbon source, xylose, by harnessing the natural galactose regulon in this organism. This is a very interesting study that includes some fundamental research on galactose metabolism, protein engineering, and metabolic engineering, and should be of interest to a wide range of researchers. The study claims that using the GAL regulon to express the catabolic enzymes for xylose utilization has some advantages over constitutive expression of the catabolic pathway. This idea has a lot of potential. However, this study has

some serious problems and unanswered questions that need to be addressed:

1) The authors should be cautious with their wording, specifically about calling this the “first design rules for incorporating a non- native nutrient assimilation system in a synthetic biological system”. What they call "design rules" are the well-known principles of the galactose inducible system in yeast, and are not new. There are also several examples of previous yeast metabolic engineering studies in which close attention was paid to the design of systems to utilize and co-utilize natural and non-natural substrates, not from the standpoint of catabolic routes, but rather strategies for substrate assimilation (xylose, arabinose, glucose, cellobiose, acetate -- see studies by Jamie Cate, Yong-Su Jin, and others). This study may represent a novel strategy in this theme, but it is an overstatement to call it "the first".

We thank the reviewer for pointing this out. Perhaps it was too presumptuous to call this system the “first”. We meant to emphasize that this is the first demonstration that genome-wide control of systems that contribute to fast growth can be activated by a non-native nutrient. In particular, our emphasis was to highlight that instead of engineering non-native nutrient assimilation in a piecemeal manner (such as would be required if XylR, was used for example), hijacking a native control method can provide a faster and more general way to achieve high growth rates.

We have re-phrased several sentences to accommodate these concerns.

The corresponding changes to the manuscript are included here:

While significant effort has been expended to elucidate design rules for anabolic metabolism, this work provides the first design guidelines to enable a global response as well as catabolism of a non-native nutrient stimulus.

However, as far as we know, this is the first known engineering effort that rationally couples nutrient sensing to direct global cellular state for fast-growth while also repressing stress and starvation responses.

2) The text contains several statements about previous studies that could be interpreted as somewhat derogatory. For example calling them or implying that they are simplistic, disregarding, or inelegant.

Thank you for this comment. Our purpose was never to diminish the significance of the work performed by other groups. We have re-worded the sentences and removed the words like “simplistic” from discussion of work by other groups. We very much appreciate the work done by the many leaders in the field of xylose metabolic engineering and benefit significantly from the results and insights from their works.

The corresponding changes to the manuscript are included here:

From:

Current efforts to engineer nutrient assimilation pathways take a **simplistic** approach of overexpressing catabolic pathway enzymes without regard for how that integrates into the larger cellular infrastructure that encompasses central metabolism, stress-responses, cell doubling, etc.

To:

Current efforts to engineer nutrient assimilation pathways take a **straightforward** approach of overexpressing catabolic pathway enzymes without regard for how that integrates into the larger cellular infrastructure that encompasses central metabolism, stress-responses, cell doubling, etc.

From:

We engineered the GAL regulon into a xylose (XYL) regulon, which resulted in better growth and final

cell density compared to simple constitutive expression of upstream xylose metabolic genes.

To:

We engineered the GAL regulon into a xylose (XYL) regulon, which resulted in better growth and final cell density compared to constitutive expression of upstream xylose metabolic genes.

3) When they compare the regulon and constitutive systems in both, galactose (natural GAL regulon) or xylose experiments, they do not address the possibility that the effects they see are caused (at least in part) by differences in promoters strength. On average, GAL1p, GAL10p, and GAL7p (in galactose) are stronger promoters than TEF1p, GPM1p, or TPI1p – it is not clear how strong these inducible and constitutive promoters are in xylose. Furthermore, the strength of GPM1p, and TPI1p promoters has only been characterized in glucose, and they may be weaker in galactose or xylose. The difference in strength between these promoters could explain at least some of the positive effect they see in utilizing the GAL system to drive galactose and xylose metabolism. Also, the claim that they use the strongest-known constitutive promoters is not true. The TDH3p (also known as GPD1p) promoter is stronger than TEF1p, GPM1p, or TPI1p, at least in glucose. Again, the strength of these promoters should be assessed in the carbons sources in which they are utilized (xylose or galactose), as they may have different strengths in different carbon sources. They need to rule out that differences in promoter strength plays a significant role in their results.

Thank you for bringing this up. While strength of promoters can be an issue, we have provided some additional data to assuage the concern that the strength of the promoter is not a major factor in determining the differences in growth rates between GAL-CONS/WT and XYL-CONS/XYL-REG pairs. In Fig S1A, we have compared the expression of EGFP driven by *GAL1p*, *GAL7p*, *GAL10p*, *TEF1p*, *GPM1p*, and *TPI1p* under galactose induction. In Fig S7A, we carried out a comparison of EGFP driven by *GAL1p*, *GAL10p*, *TEF1p* and *TPI1p* under xylose induction. Both sets of experiments were carried out to replicate the actual growth conditions. Along with these, we have also included mRNA expression data from RNA-seq analysis for the sets of genes, when the strains were grown in either galactose or xylose. In the case of growth and induction in galactose, both RNA-seq analysis as well as fluorescence based analysis agree that constitutive promoters result in a much higher expression level of GAL metabolic genes than GAL promoters. However, in the case of xylose, while fluorescence based analysis of promoter strength suggest similar strengths for all the promoters, RNA-seq analysis show that, mRNA levels of *GAL1p* and *GAL10p* promoters in XYL-REG strain to higher than constitutive promoters, TEF1p and TPIp in XYL-CONS strain. It is possible that either increased levels of mRNA observed in RNA-seq analysis doesn't translate completely into protein or presence of sucrose as the growth substrate suppresses GAL regulon resulting in lowered fluorescence of pRS426-GAL1p and pRS426-GAL10p strains. Either way, the difference in expression strengths of GAL regulated promoters and constitutive promoters alone does not explain the difference in growth rates as increased expression need not directly translate to increased growth rate. In the case of GAL-CONS, though expression of *GAL1,7,10* genes were several folds higher than in GAL-REG, growth rate was lower. Finally, as seen from RNA-seq analysis (**Fig 6**), the increased expression does not explain the observed upregulation of several growth-related pathways.

We agree with the reviewer that *TDH3p* is the strongest known promoter and is in fact stronger than *TEF1*, *TPII*, and *GPMI* in galactose too, albeit only slightly. We have removed the phrase and modified the manuscript accordingly.

The corresponding changes to the manuscript were added as supplemental notes (note 1 and note 3), which are included in here:

Main manuscript:

The decrease in growth rate can either be attributed to inability of the strain GAL-CONS to activate the

downstream genes of GAL regulon that are required for growth, or the difference in promoter strengths between the GAL and constitutive promoters that transcribe the Leloir pathway genes, or both. To determine true cause of growth rate decrease, we re-introduced Gal4p in GAL-CONS but deleted Leloir pathway genes under its native promoter as well as *GAL3* and *GRE3* (which encodes for non-specific aldose reductase) to create the strain GAL-CONS-*GAL4* (**Fig 1A**). This re-factored, partially-coupled, system should enable activation of downstream genes through Gal1p-Gal80p-Gal4p pathway³¹, but keeps the Leloir pathway genes out of the GAL regulon control. Thus, if the downstream genes of the GAL regulon assist in growth, the partially coupled strain should have growth rates higher than the GAL-CONS strain. On the other hand, if the observed decrease is a result of the difference in promoter strengths between native GAL promoters and constitutive promoters, the GAL-CONS-*GAL4* strain should have the same growth rate as that of the GAL-CONS strain. We tested the GAL-CONS-*GAL4* strain for growth in galactose, and observed that the strain recovers a significant portion of its growth fitness relative to GAL-CONS (**Fig 1B**) suggesting that the downstream genes under the control of the regulon *trans*-activated by Gal4p positively affect the ability of yeast to grow on galactose.

We would also like to point out that mRNA levels of *XYLA*3* and *XKS1* from XYL-REG strain is several folds higher than expression in XYL-CONS strain and can to some extent contribute to the observed increase in growth rate.

Supplemental Note 1: Characterization of expression strengths of GAL-activated versus constitutive promoters

To assess Gal4p mediated activation of genes that assist growth in galactose, we compared growth between strains that have Leloir genes under GAL-activated promoters (WT strain) and constitutive promoters (CONS-GAL). To rule out the possibility that the growth defect observed in CONS-GAL is due to lower expression strength of constitutive promoters, we compared the expression strengths of GAL-activated (*GAL1p*, *GAL7p* and *GAL10p*) and constitutive promoters (*TEF1p*, *TPI1p* and *GPM1p*) by placing EGFP gene downstream of these promoters. Since the WT strain has a single copy of Leloir pathway genes, we placed the *GAL_{promoter}* – EGFP constructs in pRS406 integration plasmid and knocked it into the *URA3* locus of the chromosome. To account for locus-based expression differences, we also cloned the constructs in low copy plasmids (2 – 5 copy numbers³) and considered them to be the maximum possible expression of GAL promoters in the WT strain. Next, we cloned EGFP under constitutive promoters in multicopy pRS426 plasmid and compared their fluorescence in galactose (**Fig S1a**). Comparing the results, it is clear that the constitutive promoters in high copy plasmids have stronger expression strength than single copy GAL promoters integrated in the chromosome or similar expression strength with low copy GAL promoters. This observation is consistent with mRNA expression data obtained from RNA-seq analysis (**Fig S1b**).

Supplemental Note 3: Characterization of expression strengths of *GAL1p/10p* and constitutive promoters in xylose

To check if there is a difference in expression strengths between *GAL1p/10p* and *TEF1p/TPI1p* results in observed difference in growth rates of XYL-REG and XYL-CONS strain in xylose, we transformed VEG16 strain with either pRS426-GAL1p, pRS426-GAL10p, pRS426-TEF1p or pRS426-TPI1p plasmid along with XYL regulon plasmids (pVEG16* and pVEG17*) and compared the fluorescence levels after growing the strains in sucrose along with xylose. To replicate the conditions of growth on xylose, we used high copy plasmids. Since XYL-REG and XYL-CONS strains were grown in 2 % xylose, we checked for fluorescence in 2 % xylose, as well at maximum xylose concentration of 4 %. From **Fig S7a**, it can be seen that the fluorescence at 2 % xylose concentration, constitutive promoters have 1.6-fold higher fluorescence than the GAL promoters and at 4% xylose concentration, the fluorescence levels remain the same. On the contrary, mRNA expression data from RNA-seq analysis reveal mRNA levels of *XYLA*3* and *XKS1* from XYL-REG to be several folds higher than XYL-CONS (**Fig S7b**). It is possible that either increased levels of mRNA don't translate completely into protein or presence of sucrose as the growth substrate results suppresses GAL regulon resulting in lowered fluorescence of pRS426-GAL1p and

PRS426-*GAL10p* strains. Either way, the difference in expression strengths of GAL regulated promoters and constitutive promoters alone does not explain the difference in growth rates as increased expression need not directly translate to increased growth rate. In the case of GAL-CONS, though expression of *GAL1,7,10* genes were several folds higher than in GAL-REG, growth rate was lower. Finally, as seen from RNA-seq analysis (**Fig 6**), the increased expression does not explain the observed upregulation of several growth-related pathways.

4) It is not clear why the downstream GAL genes should benefit xylose catabolism. Galactose involves the Leloir pathway while xylose involves the pentose phosphate pathway. The authors mention in the discussion that these downstream genes are not galactose-metabolism specific, but they provide no data or references for this claim. The experiments done in this study do not rule out the possibility that the differences in promoter strengths are responsible for the REG systems being superior to the constitutive promoters. They also do not offer any mechanistic explanation for how these downstream genes could enhance xylose catabolism (not even hypothetical). If downstream genes are so important for the efficient utilization specifically of galactose, as opposed to say glucose, or maltose, why should those same downstream genes benefit xylose assimilation? Conversely, if they are not specific for galactose, why are they so tightly regulated by the GAL system if they could also enhance glucose or maltose catabolism?

We thank the reviewer for this insightful comment. We also apologize for not being clear on what we mean by “downstream genes”. In our work, the Leloir pathway and PPP are considered “upstream”. Since there are only expression based evidences on the genes controlled by the GAL regulon, we refer to any genes apart from the “upstream” genes as “downstream”. We extracted the genes annotated to be regulated by the GAL regulon based on either differential expression of genes on *GAL4* downregulation or through Gal4p-DNA binding evidence, from YEASTRACT and consider them as the set of genes that could be potentially regulated either directly or indirectly by the GAL regulon. Many of these genes are not specific for a single sugar metabolism but are rather universal. It is to be noted that most of the expression based evidence is based on Hu et al., 2007 and re-analysis by Reimand et al., 2010 where *GAL4* was not knocked out, since that would result in inability of the strains to grow on galactose. Instead, *GAL4* was downregulated using *tetO* operating sites. Thus, it is possible that several of these genes might not be directly regulated by GAL regulon. It is also possible that other weakly controlled downstream GAL regulon genes might have been excluded from the analysis.

In our RNA-seq analysis (that is newly added in this revision), we have compared the REG strains and CONS strains to provide a better understanding of the genes regulated (either directly or indirectly by GAL regulon). By comparing the REG strains with CONS strains grown in the same sugar, we exclude the effects of metabolism and focus in on the effects of GAL regulon. As shown in the paper, the overlap of these two sets provided a list of 181 genes. Even from a conservative standpoint, if we assume only overlapped set of genes to be GAL regulon controlled, it can be seen that many of these genes are involved in cell wall and mitochondrial biogenesis, which are not sugar specific pathways, but are generally growth-related.

From our analysis, the gene expression profile for the REG strains are significantly different than those for the CONS strains. As can be seen from the data, that most genes are not strongly controlled by the GAL regulon. Instead, we argue that the combined effect of low levels of up-/down-regulation of multiple genes provide significant benefit to cell grown even in xylose. Thus, we believe this system is generalizable to other carbon sources as well.

References:

1. Hu, Z., Killion, P. J., & Iyer, V. R. (2007). Genetic reconstruction of a functional transcriptional regulatory network. *Nature Genetics*, 39(5), 683–687. <https://doi.org/10.1038/ng2012>
2. Reimand, J., Vaquerizas, J. M., Todd, A. E., Vilo, J., & Luscombe, N. M. (2010). Comprehensive reanalysis of transcription factor knockout expression data in *Saccharomyces cerevisiae* reveals many new targets. *Nucleic Acids Research*, 38(14), 4768–4777. <https://doi.org/10.1093/nar/gkq232>

The corresponding changes to the manuscript are included here:

A separate section on RNA-seq analysis has been added to the Results, methods, discussion, and supplemental sections. The following are the pertinent sections from Results and Discussion sections.

Results:

Characterizing regulon-assisted and conventionally engineered strains

To provide an insight on the genes differentially expressed in REG strains (GAL-REG (WT) and XYL-REG) when compared to CONS strains (GAL-CONS and XYL-CONS), that result in vastly different growth phenotypes, we carried out RNA-seq to profile their transcriptome. We used triplicates of these strains grown in their respective carbon sources, harvested them during mid-exponential phase of growth and used them for RNA-seq. We found that the transcriptome profiles of the REG and CONS strains varied drastically (**Fig S9**). Next, we carried out differential gene expression analysis between XYL-REG and XYL-CONS as well as between GAL-REG (WT) and GAL-CONS strains using the limma⁶ and edgeR⁷ packages. Genes with a statistical *p*-value less than 0.05 after Benjamini–Hochberg correction were considered as differentially regulated. Total of 4202 genes were differentially regulated between GAL-REG and GAL-CONS strains and 3314 genes between XYL-REG and XYL-CONS strains (**Fig 6a & 6b**). We reasoned that if there are genes either directly or indirectly regulated by Gal4p, they would be differentially expressed in not just the strains grown in galactose (GAL-REG versus GAL-CONS) but also in strains grown in xylose (XYL-REG versus XYL-CONS). Further, we also hypothesized that since both the CONS strains lack regulation for sugar detection, those strains should exhibit a starvation-like response. To test both, we decided to select for genes that are up- and down- regulated in both the differential gene expression analysis. We found 452 genes that were upregulated and 507 genes that were downregulated genes in the regulon strains (**Fig 6c**).

Next, we evaluated the functional relation between these genes by examining the Gene Ontology (GO) biological process terms that are enriched in up- and down- regulated genes. Of the genes that are upregulated in the REG strains, we found 36 enriched GO terms, including those relating to mitochondrial translation and transport, cell division, ATP production, protein import etc. We also found 11 GO biological process terms in the case of genes that were downregulated, which were involved in processes such as fatty acid and lipid metabolism, DNA repair, response to stimulus etc. (**Fig 6d and Fig S10**). Out of the genes enriched under the GO term for response to stimulus, 58 genes were associated with stress response. Since, we hypothesized many of these genes to be regulated by Gal4p, we extracted genes that are known to be regulated by Gal4p from YEASTRACT^{8,9} and compared them with differentially regulated genes from our analysis. 181 genes were identified as hits from the overlap of the two sets (**Fig 6e**). GO term analysis of the genes show that many of the processes previously enriched in upregulated genes such as cell division, cell wall organization, mitochondrial translation and membrane transport were enriched, suggesting that upregulation of these vital cellular pathways are a direct consequence of activating the GAL regulon. Interestingly, none of the previously enriched GO process terms from the downregulated genes were enriched in this analysis indicating that GAL regulon is not involved in downregulation of genes observed in REG strains. Between the REG and CONS strains, we also analyzed for differentially expressed genes that code for transcription factors (TFs) (**Fig 6f**). We identified 29 TFs, of which several are involved in response to stress, nutrient starvation, and DNA replication stress, such as *CBF1*¹⁰, *RPH1*^{11,12}, *GSM1*¹³, *HAA1*¹⁰, *WARI*¹⁴, *YOX1*^{10,15}, *SUT2*¹⁶, *MIG3*^{17,18} and *GCN4*^{19,20} were upregulated in the CONS strains. TFs involved in gluconeogenesis and glyoxylate cycle (*RDS2*²¹) as well as drug resistance (*RDRI*²² and *STB5*²³) were also upregulated in CONS strains. Upregulation of the above-mentioned TFs along with GO term enrichment of processes involved in DNA repair as well as upregulation of several stress response genes in the CONS strains reaffirms our hypothesis that CONS strains exhibit a starvation- and stress-like response when grown in carbon sources for which nutrient sensing systems are absent. In the case of REG strains, TFs responsible for cell wall production (*INO4*^{24,25}), cell cycle (*SWI5*^{26,27} and *HCM1*^{28,29}), and flocculation suppression (*SFLI*³⁰) were upregulated (**Fig 6e**). Taken together with growth studies, the data suggests GAL regulon controlled upregulation of

pathways involved in growth such as cell wall maintenance, cell division, mitochondrial biogenesis, and cell cycle progression. On the other hand, unregulated constitutive expression of sugar metabolizing genes seems to trigger stress, starvation, and DNA damage responses.

Fig 6: RNA-seq analysis of REG and CONS strains. Differential Expressed Genes (DEG) between (A) GAL-REG (WT) versus GAL-CONS grown on galactose and (B) XYL-REG versus XYL-CONS grown on xylose. DEG that are upregulated are shown in red while downregulated genes are shown in blue. Common DEGs with high fold change values are labeled and shown in black. (C) Number of genes that are differentially expressed between REG and CONS strains as well as genes common between the two pairs. (D) Relevant GO biological process terms of common DEGs. Heatmap of normalized log counts of DEG (E) controlled by Gal4p, and (F) that are transcription factors.

Discussion:

Transcriptome analysis by RNA-seq revealed that genes responsible for cell wall biogenesis, mitochondrial biogenesis, and ATP biosynthesis were upregulated in the REG strains. On the other hand, genes involved in response to stress, starvation, and DNA damage, and lipid metabolism were upregulated in the CONS strains as a consequence of being forced to metabolize unrecognized nutrients. Thus, the GAL regulon seems to aid in growth by upregulating several growth-related pathways and transcription factors while suppressing stress and starvation responses, which are upregulated in strains with unregulated nutrient catabolic pathways.

One way to obtain evidence that the downstream genes enhance xylose catabolism may be to induce a XYL-CONS strain growing in xylose with low concentrations of galactose (in a strain in which low levels of galactose can induce the downstream genes). If low levels of galactose can enhance the strain's growth on xylose while demonstrably inducing the downstream GAL genes, it would provide more convincing evidence to their claim that these downstream genes benefit xylose metabolism.

This is indeed an excellent suggestion. We recently conducted this experiment, but the data did not shed additional light on this question since galactose inhibits uptake of xylose. Both sugars use the same transporters and galactose, being the native nutrient, is a preferred substrate. To more directly demonstrate that XYL-REG activates the downstream growth-related genes, we are now providing more direct evidence through RNA-seq. Fig.6 clearly demonstrates that the transcriptome of the XYL-REG being significantly different from that of XYL-CONS strain. We therefore conclude that the global transcriptome remodeling in XYL-REG is responsible for its enhanced growth.

Please see response to your query #4 (above) for pertinent changes to the manuscript.

5) In lines 137 and 138, it is not clear how saturation mutagenesis of one position (109) can lead to 3000 variants. Please clarify.

We apologize for not being clear. Saturation mutagenesis with NNK codon provides a maximum diversity of 32 codons, we screened a library with ~100× more variants to ensure thorough coverage of all mutations.

The corresponding changes to the manuscript are included here:

To explore other residues at position 109 that could improve fluorescence, we carried out single site saturation mutagenesis with NNK codons to obtain a diversity of 32 codons and screened 3,000 variants (~100-fold coverage).

6) They need to explain how they measured or estimated the diversity of their error prone PCR libraries.

The mutated gene was spliced with *GAL3p* promoter and *TEF1t* terminator using OE-PCR, restriction digested, cloned into pVEG7 plasmid and electroporated in *E. coli* NEB5α cells. Five transformants were randomly chosen, plasmid was extracted, and the gene was sequenced to determine the error rate of the

library obtained.

The corresponding changes to the manuscript are included here:

The mutated gene was spliced with *GAL3p* promoter and *TEF1t* terminator using OE-PCR, restriction digested, cloned into pVEG7 plasmid and electroporated in *E. coli* NEB5 α cells. Five transformants were randomly chosen and their gene sequenced to determine the error rate of the library.

7) What are the implications of getting two distinct populations in their flow cytometry measurements as they increase xylose concentration? It seems that the genes are either completely on or off, with no intermediate level of expression. This is reported but not discussed. They must also summarize the similarity of their results with the observations of Brandman et al, rather than just mentioning consistency and referencing their study.

We apologize for not being more explicit with these discussion items. We have now edited the manuscript to better reflect this concern. The bimodal population distribution is a hallmark of bistable systems such that the cells are either in a fully ON state or fully OFF state with no intermediate levels of activation. The GAL regulon activation is a bistable system where the cells switch between ON and OFF state depending on the presence of the stimulus. We confirmed that characteristics of the bistable system, bimodality (where, in a population that is exposed to xylose, the GAL regulon in individual cells are either turned ON or OFF) and hysteresis (a history dependent response, where pre-incubation with xylose results in a higher population of cells responding to xylose than in cells that are not pre-incubated, resulting in a hysteresis shaped loop) are retained in the synthetic xylose regulon.

In 2005, Brandman et al., using mathematical simulations hypothesized that in interlinked dual positive feedback loops with a fast and slow feedback loop, results in faster response as well as stable signal output with low noise when compared to using either of the feedback loops in isolation. The fast response was attributed to the fast feedback loop and the low noise with a stable response was attributed to the slow feedback loop. Since *GAL3p* and *GAL1p* promoters act as fast and slow feedback loops, we decided to test if the absence of slow feedback loop (*GAL1p*) would result in increased noise in the system. To test that, we calculated coefficient of variation (CV), a measure of cellular heterogeneity and noise, for dual feedback and single feedback systems. Dual feedback loop consistently had lower CV than single feedback loop across different concentrations of xylose tested (**Fig 4F**), consistent with observations of Brandman et al.³¹

The corresponding changes to the manuscript are included here:

GAL regulon has been known to exhibit bimodality (results in heterogenous population in suboptimal environment, thus increasing fitness) and hysteresis (a history dependent response to galactose), which are characteristic features of a bistable system³². To test if the xylose activated regulon still retains bistability observed in the parent regulon, we decided to test bimodality and hysteresis in the XYL regulon. To demonstrate bimodality, we integrated *GALI10p-EGFP-T* cassette into the chromosome, and compared fluorescence of the two feedback systems at a cellular level under different concentrations of xylose using cytometry. Over the concentration range tested, we observed distinct populations of cells that were either turned ON or OFF confirming that the xylose regulon still retains bimodality (**Fig 4D**). Next, we pre-incubated the yeast strain carrying dual feedback system in media with and without the inducer (xylose) for 24 hours, and later shifted the cells to media with varying concentrations of xylose. We observed a pre-incubation-dependent response and at all concentrations of xylose tested. The cells that were pre-incubated with xylose showed a higher fluorescence than strains that were not incubated with xylose (**Fig 4E**). Together, these data show that the GAL-type xylose regulon retains bistability observed in the galactose regulon³². In 2005, Brandman et al., using mathematical simulations hypothesized that in interlinked dual positive feedback loops with a fast and slow feedback loop, results

in faster response as well as stable signal output with low noise when compared to using either of the feedback loops in isolation. The fast response was attributed to the fast feedback loop and the low noise with a stable response was attributed to the slow feedback loop. Since *GAL3p* and *GAL1p* promoters act as fast and slow feedback loops, we decided to test if the absence of slow feedback loop (*GAL1p*) would result in increased noise in the system. To test that, we calculated coefficient of variation (CV), a measure of cellular heterogeneity and noise, for dual feedback and single feedback systems. Dual feedback loop consistently had lower CV than single feedback loop across different concentrations of xylose tested (**Fig 4F**), consistent with observations of Brandman et al.³¹

8) The comparison of their strains to previously developed xylose-utilizing strains is a bit misleading. In their discussion, the authors compare the growth rates of their strains in complex YP medium with the growth rates of previous strains (Zhou et al, and Lee et al) obtained in SD medium. If they make the more fair comparison using the growth rate of their strains in the same medium as previous measurements (SD medium), then their XYL_CONS and XYL-REG strains are both substantially slower than other constitutive strains. This raises the question of why their constitutive strain is so slow, compared to previously developed constitutive strains? It also suggests that the XYL-REG strategy is not effective at achieving the growth rates obtained in existing (constitutive) xylose-utilizing strains.

(This again raises the possibility that their results are largely due to better expression of catabolic enzymes from the strong GAL promoters relative to the constitutive promoters they used, which may be a lot weaker than the authors assume – especially in galactose or xylose).

This is an excellent point. We have now edited the manuscript for better comparison. The higher growth rates obtained in previous studies (Zhou et al, Lee et al.) are after extensive adaptive evolution experiments, which we have not subjected our strains to (and would be beyond the scope of this work). The initial growth rates (pre-adaptive evolution) obtained in those works are comparable to that of the XYL-CONS strain constructed here. By overexpressing *XYLA*3*, *XKS1* and *TAL1* constitutively, Lee et al., 2012 were able to obtain a growth rate of 0.061 h^{-1} , which is similar to the growth rate obtained for our XYL-CONS strain. It is to be noted that in most studies, constitutively expressed strain carrying *XYLA*, *XKS1* and *TAL1* have very low growth rates ($< 0.06 \text{ h}^{-1}$) and the observed growth rate for XYL-CONS is mainly due to *XYLA*3* mutant, as shown by Lee et al., 2012. As we argue in previous points (#3, #4), the differences in promoter strength imparts minor improvements in growth rates as is also seen in the comparison between GAL-CONS and GAL-CONS-*GAL4* strain (Fig 1). We attribute the major cause to be the global remodeling of the transcriptome in XYL-REG relative to XYL-CONS as mentioned previously.

The corresponding changes to the manuscript are included here:

We engineered the GAL regulon into a xylose (XYL) regulon using minimal metabolic engineering, and obtained better growth and final cell density compared to constitutive expression of upstream xylose metabolic genes. It is to be noted that in most studies, without adaptive evolution, the growth rate of strains grown in xylose is low. The growth rate of XYL-CONS, 0.06 h^{-1} is obtained is due to superior *XYLA*3*, which was engineered by Lee et al., 2012, who report similar growth rate in their work³³. While a number of studies have obtained higher growth rates than what we have reported, these strains have all been engineered extensively with overexpression of all of the non-oxidative pentose pathways as well as adaptive evolution³⁴⁻³⁷.

9) Most of their experiments were done using 2-micron plasmids, including when developing the XYL-REG and XYL-CONS strains, and measuring the expression of other downstream genes under galactose or xylose. However, it is well known that 2-micron plasmid can produce transformants with widely heterogeneous phenotypes (not all colonies are equal). They must report how homogeneous were the

phenotypes they observed across transformants, or report how many colonies they needed to screen to find the strains they analyzed in more depth.

This is an excellent, often underappreciated point. We did not notice any major heterogeneity on our transformants. All colonies were picked at random and our results were consistent across colonies. We did not specifically choose any colonies based on any explicit criterion to analyze further.

10) The authors overstate the impact of their findings when they claim to have found an “outline for the basic design guidelines for engineering synthetic non-native regulons for nutrient assimilation”. This study is indeed very interesting, and has great potential, but it is difficult to find any general rule here. The system may have worked (provided they rule out other explanations above) because of structural similarities between xylose and galactose, the availability of the natural galactose regulon in yeast, and ultimately Gal3p’s natural ability to recognize xylose. But how are the rules they claim to have identified any different from what we already know about the GAL regulon? Also, if these rules are to be used as guidelines, how can they be extrapolated for other non-natural substrates, say formate, methanol, or cellobiose, for which GAL3 is less likely to be an effective substrate sensor?

We apologize for not being more explicit about these. We chose this as a test-case since the GAL regulon supports very high growth rates on galactose. Thus, we demonstrated that integrating non-native nutrient regulatory control into an innate system can yield significant benefits. We also wanted to highlight that regulatory control for non-native nutrients is currently not considered as a design option and we demonstrate one method in which this could provide significant benefits. With the new RNA-seq analysis, we provide evidence of why the design implementation supports superior growth relative to traditional designs.

While we chose not to engineer the GAL3p active site to accommodate xylose (due to its natural promiscuity), it may, in theory, be performed for other substrates like formate, methanol etc. It is unclear if larger substrates like cellobiose can be accommodated without more drastic changes to GAL3p’s structure. However, in theory, it is possible. We also believe if native nutrient-dependent response systems that remodel transcriptome at the genome-scale are identified, they could provide a blueprint to engineer non-native nutrient response system as well.

11) Figures 4G and S5 and the experiments they describe are not very clear. What values correspond to cells grown in sucrose (or glucose)? They do not seem to be in the figure. If they are not shown, then what is the point of the experiment? Is it not to compare levels of induction of downstream genes by xylose or galactose with respect to a baseline sugar (sucrose or glucose)?

We apologize for not being clear with the discussion of these results. In Fig 4G, all cells are grown in ethanol/glycerol mix and in Fig S4, they are grown in sucrose. The aim of the experiment was to demonstrate that in the presence of xylose or galactose how downstream genes activated by regulon would vary between REG and CONS strains. The control strains contain no xylose or galactose responsive systems and represent the absence of regulon condition in the CONS strains. The other strains contain the XYL- or GAL-inducible systems (but not xylose and galactose catabolic genes) and represent the REG strains. We wanted to demonstrate that two sugar-responsive system behave very similarly to each other and activate expression of genes that are much further downstream of PPP/Leloir pathway and involved in general metabolism and growth. Explicitly, these genes are not strongly regulated by either system, but demonstrate that the regulon is a genome-scale system to remodel growth and metabolism. We have provided additional evidence of this using RNA-seq. **Fig 6** shows explicitly how the regulons re-shape transcriptome relative to that without regulon.

The corresponding changes to the manuscript are included here:

Caption for Fig 4G:

Promoters that drive expression the downstream genes of the galactose regulon were used to express EGFP. Fluorescence was measured for cells grown on ethanol/glycerol in the presence or absence of xylose and galactose regulon.

12) They need to clearly define all the variables and constants in their models, as well as list all their assumptions (some assumptions are mentioned in the methods section, but it is not clear that this is a complete list).

We have included a supplemental note (Supp note. 2) with more details on modeling as well as added the list of constants used in the supplemental section of the paper.

13) Color code is missing for Fig S2.

Fixed.

14) There is an author's comment left in Fig S6

Fixed.

15) Typo in line 230: "include" is written twice

Comments 13-15 have been addressed and necessary changes have been made in the manuscript.

Reviewers' comments:

Reviewer #1 (Remarks to the Author):

In the revision, Gopinarayana and Nair scale back some claims about novelty and present new RNA-seq data. Although the manuscript has improved in places, there are issues that have come out in the revision and response letter that were not apparent in the original draft that require attention.

In my original point 2 I questioned the validity of comparing the constitutive promoter designs against the synthetic system. The authors raise a valid point that both designs could be improved with fine tuning, not just the constitutive system and I agree. However, in their response they point out the following: "The benefits of the feedback system are minor. The major benefit is due to GAL4-based activation of downstream genes." This is in contrast with the weight that the feedback design is given in the manuscript (all of Fig. 4, modeling discussion and results, significant figures in supplementary). If it is the case that the feedback aspects of the design are of relatively minor importance and that just activating the downstream genes in response to xylose is what is necessary this would significantly simplify the design. This would also help address point 3 from my original review where I questioned the generality and utility of the findings.

My recommendation is that the authors revisit their design, focusing instead of regulation of downstream genes and not recapitulating the feedback architecture of the Gal network, which introduces complexity without significant benefits. This would mean generating new data for Figs. 5 and 6 and reworking the text of the manuscript to refocus on the parts of the design that actually matter. I appreciate that this is a non-trivial request, but it would increase the ultimate impact of the work.

Other issues:

- * There are many grammatical errors and I recommend the manuscript be reread carefully to address these (e.g. Line 93: "lack of evidence that demonstrate", Line 152: "Fluorescence profile ... showed that", Line 158: "three variants with better fluorescence profile", Line 198: "while retaining a low level basal expression", and many more)
- * Line 71: Says the Leloir pathway genes were deleted, but I believe they are actually placed under the control of a constitutive promoter.
- * Line 148: remove ":"
- * Line 154: "approximately 0.5 OD of the cells" Please revise. OD is not a volume.
- * Please clarify what "individual triplicates" are? Are these biological or technical replicates? I am assuming the former.
- * Fig 1A: space missing before (WT)
- * Line 837: comma used in one place and dash in the next for ranges of sugar concentration
- * Fig 4D: has illegible y axis label
- * Fig 6E and F: have illegible labels on right hand side of figures

Reviewer #2 (Remarks to the Author):

The authors make a good effort to address most of the issues raised by the reviewers with either new experimental data or clarifications in the text.

Unfortunately, the authors disregard concern #10 of reviewer #2, and continue to overstate the findings and significance of their work. They insist that they have discovered (or developed) some new "design rules" that can be used as "guidelines" to engineer regulon-based non-native nutrient assimilation. These guidelines or rules are never formally listed, or shown specifically why they are universal or useful for other nutrients. This claim comes across as an attempt to rebrand what has

already been known about the GAL-regulon for decades regarding nutrient sensing (input), signal transduction (computation), and global genetic regulation (output). It is very difficult to see how this strategy could be applied to other carbon sources not easily recognized by Gal3p, let alone using it as a guideline for designing non-native nutrient assimilation in other organisms. If they want to claim some universal rules they need to show that they work for at least a second nutrient and a second organism. This is very unfortunate because harnessing the GAL regulon to enhance xylose utilization in yeast is a brilliant idea that seems to work very well, and the study could well be published in Nature Communications, even if it does not reveal any universal design rules. Because of these far overreaching and misleading claims, this reviewer cannot support publishing this manuscript.

Other areas of concern remain:

- 1) While the authors address original concern #2 of reviewer 2 in the main text, the abstract of the revised manuscript still contains unnecessarily demeaning language to describe previous work: "...severely underdeveloped and simplistic constitutive expression systems are widely used to engineer catabolism."
- 2) The new experimental data on the question of promoter strength is appreciated, but still falls short. The new experiments show that the Leloir genes in the constitutive strains are indeed overexpressed (the levels are higher than the wild type when induced with galactose). The authors then assume that this gene overexpression is not detrimental to cell growth, when in fact it is well known that gene overexpression can also lead to growth defects. This potential disadvantage of the constitutive strains is not measured (for example by putting the Leloir genes under their native GAL promoters in high copy plasmids) or even discussed in the manuscript.
- 3) They did not fully address the concern of reviewer 1 regarding experimental replicates. There are a number of experiments that are still carried out with only duplicates (shown in supplementary figures).

Reviewers' comments:

First, we would like to thank both the reviewers for their comments and suggestions, which have immensely helped improve this manuscript.

Reviewer #1 (Remarks to the Author):

In the revision, Gopinayanan and Nair scale back some claims about novelty and present new RNA-seq data. Although the manuscript has improved in places, there are issues that have come out in the revision and response letter that were not apparent in the original draft that require attention.

In my original point 2 I questioned the validity of comparing the constitutive promoter designs against the synthetic system. The authors raise a valid point that both designs could be improved with fine tuning, not just the constitutive system and I agree. However, in their response they point out the following: “The benefits of the feedback system are minor. The major benefit is due to GAL4-based activation of downstream genes.” This is in contrast with the weight that the feedback design is given in the manuscript (all of Fig. 4, modeling discussion and results, significant figures in supplementary). If it is the case that the feedback aspects of the design are of relatively minor importance and that just activating the downstream genes in response to xylose is what is necessary this would significantly simplify the design. This would also help address point 3 from my original review where I questioned the generality and utility of the findings.

My recommendation is that the authors revisit their design, focusing instead of regulation of downstream genes and not recapitulating the feedback architecture of the Gal network, which introduces complexity without significant benefits. This would mean generating new data for Figs. 5 and 6 and reworking the text of the manuscript to refocus on the parts of the design that actually matter. I appreciate that this is a non-trivial request, but it would increase the ultimate impact of the work.

The reviewer makes a very valid point based on our correspondence that the major benefits of the feedback system are due to Gal4p-based activation of downstream genes. However, we would like to add that benefits of the dual feedback system are not negligible and should not be regarded as minor. If our previous correspondence indicated as such, we apologize for confusing the matter. For example, inclusion of a dual feedback system results in ~36% increase in growth rate in galactose ($\mu = 0.30 \text{ h}^{-1}$ for WT vs 0.22 h^{-1} for CONS-GAL-GAL4). To demonstrate that significant benefit is conferred to xylose as well, we have carried out growth on xylose using two new regulon designs, a single feedback regulon and a constitutively turned ON regulon (**Fig S8**). As can be seen from the new data that we have provided in the results and supplementary sections, both the designs fail to reach the growth rate and final biomass of the dual feedback system (XYL-REG strain), even though they are superior to the traditional design (XYL-CONS). We would like to emphasize that we have shown here, with more data now, that the regulatory design implementation can directly and significantly affect growth rate and final biomass density without application of evolutionary/combinatorial techniques. This has not been previously considered as a design strategy for non-native nutrient assimilation and highlights the novelty of this work and the weight put on tuning and characterizing the dual feedback control.

In the case of both GAL regulon and XYL regulon, the decrease in growth rate when feedback is absent is significant. We would also like to point out that the design and implementation of the dual feedback is relatively simple where the same gene is just expressed under two promoters of differing strength. Further, we would also like to stress that the final growth rate obtained in the case of XYL-REG strain is the synergistic effect of all the different strategies employed such as protein engineering of Gal3p, inclusion of the dual feedback design and activation of downstream genes of the regulon along with expression of xylose metabolic genes. Finally, we would like to thank the reviewer for pointing it out as it prompted to test the effect of growth on other regulon designs. From the experiments, it is clear that disruption of any of the three strategies used would result in a significant decrease in growth rate on xylose.

The corresponding changes to the manuscript are included here:

We also compared the effect of other GAL regulon design on growth in xylose. While we have shown that dual positive feedback exhibits better sensitivity and lower noise when compared to single positive feedback design, we also tested if these characteristics would also translate to improved growth. We transformed plasmids carrying necessary genes for xylose metabolism under *GAL1p* and *GAL10p* promoters along with Gal3p-Syn4.1 under *GAL3p* promoter to create a single feedback strain, XYL-REG^{SF}. In this strain, plasmid carrying Gal3p-Syn4.1 under *GAL1p* promoter necessary for dual feedback was excluded. We also tested the effect of constitutively expressing Gal3p-Syn4.1 by placing the gene under *TEF1p* promoter (XYL-REG^C). Both the strains have a growth rate of 0.12 h⁻¹ and a final OD of ~8, a 20% decrease in growth rate and a 27% decrease in final biomass, respectively, compared to the dual feedback design (Fig S8B). This clearly showcases the growth benefits of the wild-type-like dual positive feedback system in XYL-REG.

Figure S8: Growth of engineered strains in xylose. (A) Growth of XYL-REG ($\mu = 0.12 \text{ h}^{-1}$, $\text{OD}_{\text{max}} \approx 3.2$) and XYL-CONS ($\mu = 0.07 \text{ h}^{-1}$, $\text{OD}_{\text{max}} \approx 2.1$) strains in minimal medium supplemented with 2 % xylose. (B) Schematic of the regulon designs (left) and comparison of growth curves (right) in complex medium (YPA + 2% xylose) of dual feedback strain XYL-REG ($\mu = 0.15 \text{ h}^{-1}$, $\text{OD}_{\text{max}} \approx 11$), with single feedback, XYL-REG^{SF} ($\mu = 0.12 \text{ h}^{-1}$, $\text{OD}_{\text{max}} \approx 8$), and constitutively active XYL regulon XYL-REG^C ($\mu = 0.12 \text{ h}^{-1}$, $\text{OD}_{\text{max}} \approx 8$). Each data point represents average of biological triplicates \pm sd.

Other issues:

* There are many grammatical errors and I recommend the manuscript be reread carefully to address these (e.g. Line 93: “lack of evidence that demonstrate”, Line 152: “Fluorescence profile ... showed that”, Line 158: “three variants with better fluorescence profile”, Line 198: “while retaining a low level basal expression”, and many more)

We would like to thank the reviewer for carefully going through the paper. We have edited the paper for grammatical errors to the best of our knowledge and have also corrected all the mistakes the reviewer has pointed out.

* Line 71: Says the Leloir pathway genes were deleted, but I believe they are actually placed under the control of a constitutive promoter.

Thanks for pointing that out. The Leloir pathway genes were indeed deleted from the genome and then copies were placed under the control of constitutive promoters.

The corresponding changes to the manuscript are included here:

To determine true cause of growth rate decrease, we re-introduced Gal4p in GAL-CONS but deleted genomic Leloir pathway genes (and placed the genes under constitutive expression) as well as *GAL3* and *GRE3* (which encodes for non-specific aldose reductase) to create the strain GAL-CONS-GAL4.

* Line 148: remove “:”

Done

* Line 154: “approximately 0.5 OD of the cells” Please revise. OD is not a volume.

The corresponding changes to the manuscript are included here:

Triplicates of strains WT (GAL-REG), XYL-REG, GAL-CONS and XYL-CONS were grown in their respective carbon source (galactose or xylose) till mid exponential phase and approximately 2×10^7 cells were washed twice in water, pelleted and stored at -80 °C.

* Please clarify what “individual triplicates” are? Are these biological or technical replicates? I am assuming the former.

All the experiments were conducted using biological triplicates. We apologize for the confusion and have modified it in the paper wherever necessary.

* Fig 1A: space missing before (WT)

Fixed

* Line 837: comma used in one place and dash in the next for ranges of sugar concentration

The corresponding changes to the manuscript are included here:

(B) Gal3p-WT interaction with galactose and xylose measured using the fluorescence assay. VEG16 transformed with selection and screening construct was grown in sucrose with varying concentrations of galactose (2 % - 2×10^{-6} %) or xylose (8% - 2×10^{-3} %).

* Fig 4D: has illegible y axis label

Fixed

* Fig 6E and F: have illegible labels on right hand side of figures

Unfortunately, there are 180 labels for Fig 6E and 30 for Fig 6F and is hard to make them legible. The intention of these figures is to show that between REG and CONS strains there are differences in expression in either genes known to have been regulated by GAL4 or in transcription factors. Hence, we have provided the names of these genes separately in supplemental data for the reader.

The corresponding change to the manuscript Fig 6 legend is included here:
Identities of all genes in the heatmaps are provided in supplemental data.

Reviewer #2 (Remarks to the Author):

The authors make a good effort to address most of the issues raised by the reviewers with either new experimental data or clarifications in the text.

Unfortunately, the authors disregard concern #10 of reviewer #2, and continue to overstate the findings and significance of their work. They insist that they have discovered (or developed) some new “design rules” that can be used as “guidelines” to engineer regulon-based non-native nutrient assimilation. These guidelines or rules are never formally listed, or shown specifically why they are universal or useful for other nutrients. This claim comes across as an attempt to rebrand what has already been known about the GAL-regulon for decades regarding nutrient sensing (input), signal transduction (computation), and global genetic regulation (output). It is very difficult to see how this strategy could be applied to other carbon sources not easily recognized by Gal3p, let alone using it as a guideline for designing non-native nutrient assimilation in other organisms. If they want to claim some universal rules they need to show that they work for at least a second nutrient and a second organism. This is very unfortunate because harnessing the GAL regulon to enhance xylose utilization in yeast is a brilliant idea that seems to work very well, and the study could well be published in Nature Communications, even if it does not reveal any universal design rules. Because of these far overreaching and misleading claims, this reviewer cannot support publishing this manuscript.

We agree with the reviewer that the rules haven’t been used for a second non-native nutrient or for a second organism in this manuscript. While we are currently working on extending it for other sugars that are not easily recognizable by wildtype Gal3p, the work is in progress and we cannot show that data in this manuscript. Further, that is also beyond the scope of the work described in this manuscript. Our idea was never to be disingenuous in representing the outcome and impact of our work. The “design rules” described here is a methodology to incorporate non-native nutrient metabolism into native host functions by appropriating a native regulon. While we show this for only for xylose metabolism in *S. cerevisiae* in this manuscript, we posit that such semi-synthetic regulon implementations should be considered whenever non-native nutrients are to be metabolized in a model host. Of course, the implementations for different hosts may be different since their native regulatory infrastructures may be different. Our implication was never to indicate that the GAL system as a “plug-and-play” for all organisms. Further universality across multiple organisms and/or pathways is rarely shown when claiming “design rules” for regulatory control, even for RNA-based systems. However, we will concede to the reviewer that the term “design rules” could mean different things to different readers and our conclusions may qualify more as a “design strategy” rather than “design rules”. Hence, we have renamed the paper to “A semi-synthetic regulon enables rapid growth of yeast on xylose” and have made necessary editions in the abstract, introduction and discussion section of the paper to better reflect the idea that re-engineering the GAL regulon enables better growth on xylose.

Other areas of concern remain:

1) While the authors address original concern #2 of reviewer 2 in the main text, the abstract of the revised manuscript still contains unnecessarily demeaning language to describe previous work: “...severely underdeveloped and simplistic constitutive expression systems are widely used to engineer catabolism.”

We apologize for failing to modify the text in the abstract of the paper. We have removed the phrase and have modified the abstract.

2) The new experimental data on the question of promoter strength is appreciated, but still falls short. The new experiments show that the Leloir genes in the constitutive strains are indeed overexpressed (the levels are higher than the wild type when induced with galactose). The authors then assume that this gene overexpression is not detrimental to cell growth, when in fact it is well known that gene overexpression can also lead to growth defects. This potential disadvantage of the constitutive strains is not measured (for example by putting the Leloir genes under their native GAL promoters in high copy plasmids) or even discussed in the manuscript.

We agree with the reviewer that overexpression can be detrimental to cell growth. In fact, in the original draft of the paper, we had assumed that GAL-CONS has higher expression of Leloir pathway genes and that had led to slower

growth rate of GAL-ONS on galactose. Hence, we had created the control strain GAL-CONS-*GAL4* to account for the difference in expression strength. Irrespective of whether the GAL-CONS strain has higher or lower expression strength of Leloir pathway genes when compared to GAL-REG (WT), the GAL-CONS-*GAL4* strain has the **same** expression strength as GAL-CONS since both strains have the same plasmid (pCONS-GAL) expressing the Leloir pathway genes under constitutive promoters. Yet, they demonstrate very different growth phenotypes. Therefore, we stand by our claim that even though there may be some effects due to differences in Leloir pathway gene expression levels, a significant portion of the growth benefit is due to Gal4p-mediated regulatory control of downstream genes under the GAL regulon ($\mu = 0.22 \text{ h}^{-1}$ for GAL-CONS-*GAL4* vs. 0.06 h^{-1} for GAL-CONS). We would like to highlight again, that this ~4-fold difference in growth rate is observed when the exact same plasmid is used to express the Leloir pathway genes when the expression levels are expected to be nearly identical. The difference in growth phenotype between WT (GAL-REG) and GAL-CONS-*GAL4* ($\mu = 0.30 \text{ h}^{-1}$ vs. 0.22 h^{-1}) could be due to a combination of promoter strengths between constitutive (*TEF1p*, *GPM1p*, etc. vs. *GAL1p*, *GAL10p*, etc.) and/or the dynamic dual positive feedback control. Therefore, we hope that the reviewer's concerns are fully- and well-addressed.

3) They did not fully address the concern of reviewer 1 regarding experimental replicates. There are a number of experiments that are still carried out with only duplicates (shown in supplementary figures).

We have provided biological triplicate data for all the figures in the supplemental section as well in the current revision of the manuscript.

REVIEWERS' COMMENTS:

Reviewer #1 (Remarks to the Author):

In the latest revision, Gopinarayanan and Nair continue to scale back claims about novelty and clarify the benefits of the dual positive feedback loop.

1. I still have residual concerns that I listed in my original review about the potential utility and generality of the findings. As I understand it, to use this method you would have to work with a carbon source that is already at least somewhat recognized by Gal3p, then use a combination of targeted and mutagenesis strategies to mutate Gal3p to respond to the new sugar, then place the genes for growth of that new carbon source under Gal4p control, and also design a second promoter, controlled differently from the first, that regulates mutant Gal3p expression. I do think it's interesting that it works, but it's a relatively complex process, hence my concerns about broad applicability.

2. Although the growth rate data in Fig. S8 do provide evidence of potential benefits of the dual feedback strategy over the single feedback strategy it is not clear why some experiments were conducted in minimal medium and some in complex medium. Are the results consistent if the same medium is used? And are the results media dependent? (As an additional point, the authors should define the minimal media. Is this the SC medium mentioned in the Methods or something else?)

3. The manuscript is still littered with grammatical errors. This is especially true in the sections with new text. There are too many to list here, but as an example from one paragraph: line 289: "compared to single positive", line 292: "under Gal3p promoter", line 293: "In this strain, plasmid", line 293: "under GAL1p promoter", etc. The errors are not limited to the new sections.

4. Reviewer #2 made an excellent point about the authors using demeaning language and they have mostly cleared this up in the revisions, but one remaining place is line 419: "more elegant approach than prevailing strategies". This is subjective.

Reviewer #2 (Remarks to the Author):

The latest version of the manuscript satisfies the reviewers' concerns, which makes it acceptable for publication

Reviewer #1 (Remarks to the Author):

In the latest revision, Gopinarayanan and Nair continue to scale back claims about novelty and clarify the benefits of the dual positive feedback loop.

1. I still have residual concerns that I listed in my original review about the potential utility and generality of the findings. As I understand it, to use this method you would have to work with a carbon source that is already at least somewhat recognized by Gal3p, then use a combination of targeted and mutagenesis strategies to mutate Gal3p to respond to the new sugar, then place the genes for growth of that new carbon source under Gal4p control, and also design a second promoter, controlled differently from the first, that regulates mutant Gal3p expression. I do think it's interesting that it works, but it's a relatively complex process, hence my concerns about broad applicability.

The author makes a valid point regarding the utility and generality of the findings. As we had previously responded to reviewer #2, we are currently working on extending it for other sugars that are not easily recognizable by wildtype Gal3p. The work is in progress and we cannot show that data in this manuscript. Further, that is also beyond the scope of the work described in this manuscript. Based on the comment of reviewer #2, the title and the scope of the paper is now changed to focus solely on xylose. We would like to add that the focus of this work was not only to develop a method to enable rapid growth on xylose, but to also demonstrate why currently design strategies (constitutive expression) have not had more significant success in enabling rapid growth on non-native carbon-sources like xylose. Those results and insights, we posit, are expected to be broadly applicable to other nutrients and organisms, and thus of interest to a wide audience. While, it turned out that Gal3p had weak interactions with xylose we do not think that is a requirement. Engineering of Gal3p to recognize other structurally different sugars might involve several rounds of mutagenesis. We do think it possible, especially with the established high-throughput screen and selection described in this work. Further, the results demonstrating growth effects on single- and dual-feedback designs may be broadly applicable to other carbon-sources as well.

2. Although the growth rate data in Fig. S8 do provide evidence of potential benefits of the dual feedback strategy over the single feedback strategy it is not clear why some experiments were conducted in minimal medium and some in complex medium. Are the results consistent if the same medium is used? And are the results media dependent? (As an additional point, the authors should define the minimal media. Is this the SC medium mentioned in the Methods or something else?)

When the focus was on gene expression analysis and maximizing sugar utilization, we chose to use complex medium. This is because in the minimal medium we used, we did not observe complete sugar consumption even in native sugars such as glucose, galactose or sucrose, possibly due to limitation of other nutrients. Conversely, when differences in growth rates were compared, we used minimal SC medium. We do not have any reason to believe that the trends observed are media dependent. The SC medium mentioned in the Methods is the minimal medium that we have used.

The corresponding changes to the manuscript are included here:

Initial growth studies on synthetic defined media with xylose resulted in a growth rate of 0.12 h^{-1} , when compared with XYL-CONS that grew at 0.07 h^{-1} (**Supplementary Fig. 8A**). However, xylose was not fully consumed possibly due to nutrient limitation in minimal SC medium. Hence, we tested growth on complex YP medium supplemented with 2 % xylose and observed growth rates of 0.15 h^{-1} for XYL-REG and 0.06 h^{-1} for XYL-CONS.

3. The manuscript is still littered with grammatical errors. This is especially true in the sections with new text. There are too many to list here, but as an example from one paragraph: line 289: “compared to single positive”, line 292: “under Gal3p promoter”, line 293: “In this strain, plasmid”, line 293: “under GAL1p promoter”, etc. The errors are not limited to the new sections.

We have edited the manuscript for improved readability to the best of our ability.

4. Reviewer #2 made an excellent point about the authors using demeaning language and they have mostly cleared this up in the revisions, but one remaining place is line 419: “more elegant approach than prevailing strategies”. This is subjective.

We have modified the line accordingly in the manuscript.

This approach can be easily extended for other abundant, but non-native nutrients. We suggest that the regulon engineering strategy is a rational and potentially faster, and potentially more elegant approach than prevailing strategies.